# Numerion: A Multi-Hypercomplex Model for Time Series Forecasting

**Hanzhong Cao[1, a], Wenbo Yan[1,b,c] & Ying Tan[b,d,e,*]**
[a]School of Electronics Engineering and Computer Science, Peking University, Beijing
[b]School of Intelligence Science and Technology, Peking University, Beijing
[c]Computational Intelligence Laboratory, Peking University, Beijing
[d]Institute for Artificial Intelligence, Peking University, Beijing
[e]State Key Laboratory of General Artificial Intelligence, Peking University, Beijing
{2200018929, wenboyan }@stu.pku.edu.cn, ytan@pku.edu.cn

## Abstract

Many methods aim to enhance time series forecasting by decomposing the series through intricate model structures and prior knowledge, yet they are inevitably limited by computational complexity and the robustness of the assumptions. Our research uncovers that in the complex domain and higher-order hypercomplex spaces, the characteristic frequencies of time series naturally decrease. Leveraging this insight, we propose Numerion, a time series forecasting model based on multiple hypercomplex spaces. Specifically, grounded in theoretical support, we generalize linear layers and activation functions to hypercomplex spaces of arbitrary power-of-two dimensions and introduce a novel Real-Hypercomplex-Real Domain Multi-Layer Perceptron (RHR-MLP) architecture. Numerion utilizes multiple RHR-MLPs to map time series into hypercomplex spaces of varying dimensions, naturally decomposing and independently modeling the series, and adaptively fuses the latent patterns exhibited in different spaces through a dynamic fusion mechanism. Experiments validate the model's performance, achieving state-of-the-art results on multiple public datasets. Visualizations and quantitative analyses comprehensively demonstrate the ability of multi-dimensional RHR-MLPs to naturally decompose time series and reveal the tendency of higher-dimensional hypercomplex spaces to capture lower-frequency features.

## 1 Introduction

Time series prediction is crucial in fields like power, finance, transportation, and industry, with a focus on uncovering temporal patterns for accurate forecasting. Recent advancements in deep learning, including CNN-based (e.g., TCNBai et al. (2018), SCINetLiu et al. (2022)), RNN-based (e.g., LSTNetLai et al. (2018), TPA-LSTMShih et al. (2019)), Transformer-based (e.g., InformerZhou et al. (2021), AutoformerWu et al. (2021), PatchTSTNie et al. (2022), iTransformerLiu et al. (2023b)), and MLP-based methods (e.g., N-BeatsOreshkin et al. (2019), DLinearZeng et al. (2023)), have significantly enhanced this domain.

In cutting-edge solutions, techniques like sequence decomposition and multi-period pattern recognition are integrated into models, often through complex structures or manual rules, increasing computational demands and model complexity. Alternatively, some models achieve these tasks in the frequency domain with simpler architectures. For instance, SCINetLiu et al. (2022) and FiLMZhou et al. (2022a) use frequency domain decomposition, FEDformerZhou et al. (2022b) and FreTSYi et al. (2023) capture multi-period patterns, and FilterNetYi et al. (2024) learns amplitude relationships via filters. This stems from the natural divergence of time series properties in complex spaces compared to real domains, prompting the question: **Can time series exhibit additional properties in higher-dimensional hypercomplex spaces, enabling modeling through simple transformations rather than intricate structures?**

Inspired by hypercomplex number theory (Appendix L), our experiments reveal that mapping time series from the real domain to higher-dimensional hypercomplex spaces (e.g., complex numbers,

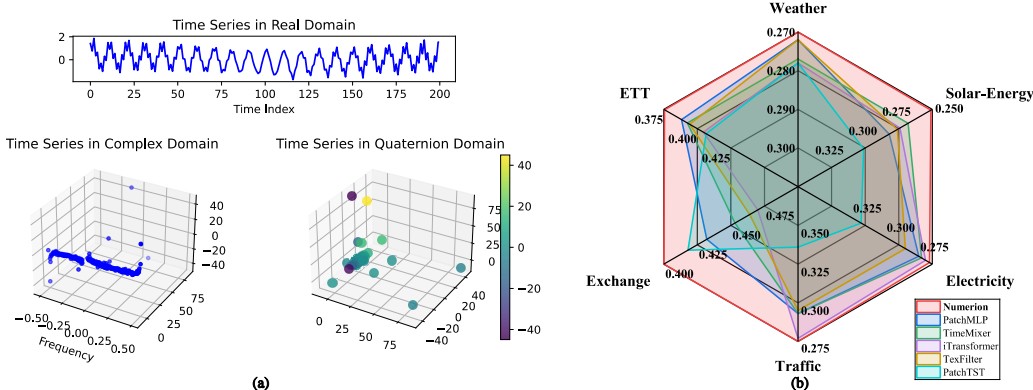

Figure 1: (a) Time Series in Real, Complex, and Quaternion Domains. (b) Performance of Numerion.

quaternions) naturally alters their characteristics, favoring lower-frequency features. As illustrated in Figure 1, mapping to the complex domain eliminates high-frequency fluctuations, unveiling a hierarchical spatial structure. In quaternion space, high-frequency features further diminish, resulting in a clustering phenomenon where nearly all data points align with the same low-frequency feature.

Based on our experimental findings, we propose that mapping time series into hypercomplex spaces of varying dimensions inherently reveals distinct latent features, eliminating the need for complex model structures. To this end, we introduce **Numerion**, a Multi-Hypercomplex time series forecasting model. Specifically, we generalize linear transformations and tanh activation functions from the real domain to hypercomplex spaces of arbitrary power-of-two dimensions, establishing a unified framework. We then design a novel Real-Hypercomplex-Real domain Multi-Layer Perceptron (RHR-MLP) to model linear and nonlinear relationships in hypercomplex spaces. By mapping time series into multiple hypercomplex spaces, we independently model temporal features using the simple RHR-MLP structure. Finally, to better integrate information from different hypercomplex spaces, we designed the Multi-Hypercomplex Adaptive Fusion mechanism to adaptively fuse them. Experiments show our method achieves state-of-the-art performance across multiple datasets. Ablation studies, visualizations, and quantitative analyses confirm that hypercomplex spaces naturally enable multi-frequency decomposition of time series, uncovering diverse latent temporal patterns, with higher-dimensional spaces tending to model low-frequency features. Our contributions are summarized as follows:

- We generalize the linear layer and Tanh activation function to hypercomplex spaces, supported by rigorous theoretical foundations, and introduce a novel Real-Hypercomplex-Real domain Multi-Layer Perceptron (RHR-MLP) architecture.

- We propose **Numerion**, a multi-hypercomplex space time series prediction model that maps time series into different hypercomplex spaces to naturally achieve multi-frequency decomposition, model distinct temporal patterns, and adaptively fuse them.

- Experiments validate the effectiveness of our method. Visualizations and quantitative analyses demonstrate that hypercomplex spaces naturally enable multi-frequency decomposition and reveal their underlying decomposition patterns.

## 2 PRELIMINARY

### 2.1 DEFINITION OF LONG-TERM TIME SERIES PREDICTION PROBLEM

The long-term time series prediction problem involves forecasting the values of target variables $\{x_{T+1}, \cdots, x_{T+P} | x_i \in \mathbb{R}^F\}$ over a future time period, utilizing historical time series data $\{x_1, \cdots, x_T | x_i \in \mathbb{R}^F\}$. Here, $T$ represents the length of the historical sequence, $P$ denotes the

---

[1] The first author and second author contribute equally to this work.

prediction horizon, and $F$ indicates the number of features in the time series. The input is represented as $\mathbf{X} \in \mathbb{R}^{T \times F}$, and the label is denoted as $\mathbf{Y} \in \mathbb{R}^{P \times F}$.

## 2.2 Definition of Hypercomplex Numbers

Hypercomplex numbers are an extension of complex numbers, generalizing their properties to higher-dimensional spaces through the introduction of additional dimensions or more intricate algebraic structures.

**Lemma 2.1** *Cayley-Dickson Algebra System: Let $A$ be an algebraic system equipped with a conjugation operation (denoted as $*$), where elements are of the form $(\alpha, \beta)$. Through iteration, a new algebraic system $A'$ can be constructed, adhering to the following rules:*

$(\alpha_1, \beta_1) + (\alpha_2, \beta_2) = (\alpha_1 + \alpha_2, \beta_1 + \beta_2); \quad (\alpha_1, \beta_1) \times (\alpha_2, \beta_2) = (\alpha_1 \alpha_2 - \beta_2^* \beta_1, \ \beta_2 \alpha_1 + \beta_1 \alpha_2^*); \ (\alpha_1, \beta_1)^* = (\alpha_1^*, -\beta_1)$

By Lemma 2.1, we can begin with the real numbers and iteratively construct algebras of any power-of-two dimension, such as the complex numbers (2D), quaternions (4D), octonions (8D), sedenions (16D), and beyond. These algebras comprise a real part and multiple imaginary units, adhering to specific multiplication rules. The properties of real numbers manifest distinct behaviors within these higher-dimensional spaces.

We define the n-dimensional hypercomplex numbers uniformly represented as:

$$c^{(n)} := a_0 + a_1 \mathbf{i}_1 + ... + a_{n-1} \mathbf{i}_{n-1}$$

where $a_0$ is the real part coefficient, $\{a_1, \ldots, a_{n-1}\}$ are the imaginary part coefficients, and $\{\mathbf{i}_1, \ldots, \mathbf{i}_{n-1}\}$ are the imaginary units. Since hypercomplex numbers are generated iteratively, the dimension $n$ must be a power of two (e.g., 1, 2, 4, ...). When $n = 0$, $c^{(0)}$ represents a real number.

To optimize these parameters within neural architectures, Numerion is rigorously grounded in the Generalized Hamilton-Real (GHR) Calculus framework. As dictated by Liouville's Theorem in hypercomplex analysis, strict differentiability is mathematically incompatible with the bounded nonlinear activation functions necessary for deep learning. GHR Calculus bridges this gap, establishing a formal mathematical basis for optimizing real-valued loss functions with respect to hypercomplex parameters while ensuring the gradient flow remains strictly constrained by hypercomplex algebraic rules. Comprehensive theoretical details and derivations regarding these gradient differentiation mechanisms are provided in Appendix H.2.

Similarly, in this paper, all vectors, matrices, and modules in hypercomplex space are denoted with a superscript $(n)$ to indicate their dimensionality. The absence of this superscript implies $n = 1$, which corresponds to the real number domain.

## 3 Method

In this section, we first generalize the linear layer and Tanh activation function to hypercomplex spaces, proposing a real-hypercomplex-real domain multi-layer perceptron (RHR-MLP) architecture. This architecture maps inputs to arbitrary hypercomplex spaces, uncovering unique features in high-dimensional spaces. Building on this, we introduce a Multi-Hypercomplex time-series prediction model, named **Numerion**, which achieves natural multi-frequency decomposition, modeling, and fusion of time series through RHR-MLP across multiple hypercomplex spaces.

## 3.1 Real-Hypercomplex-Real domain Multi-Layer Perceptron

Firstly, we define the mapping principles and generalize the linear layers and Tanh activation functions to the hypercomplex space. Subsequently, we propose a Real-Hypercomplex-Real domain Multi-Layer Perceptron (RHR-MLP) architecture, designed to model both linear and nonlinear relationships within the hypercomplex space.

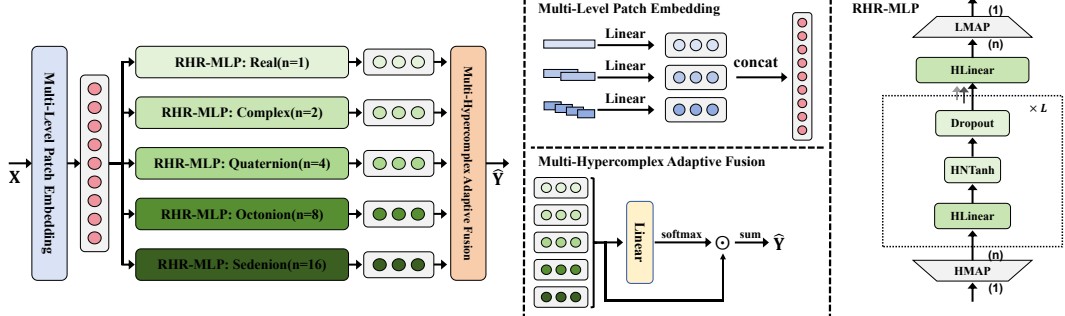

Figure 2: Overall Structure of Numerion. Primarily includes Multi-Level Patch Embedding, Multi-Dimensional RHR-MLP, and Multi-Hypercomplex Adaptive Fusion.

### 3.1.1 HIGH-DIMENSIONAL HYPERCOMPLEX MAPPING

We first define the mapping method from lower-dimensional hypercomplex numbers to higher-dimensional ones. Since higher-dimensional hypercomplex numbers are derived from lower-dimensional ones, for any $n_l$-dimensional hypercomplex number, when mapping to a higher dimension $n_h$ (where $n_h > n_l$), we adopt the principle of preserving the lower-dimensional coefficients and padding the higher-dimensional ones with zeros. This process is denoted as $\mathbf{X}^{(n_h)} = \text{HMAP}^{(n_l \rightarrow n_h)}(\mathbf{X}^{(n_l)})$.

$$
\begin{aligned}
c^{(n_l)} &:= a_0 + a_1 \mathbf{i}_1 + ... + a_{n_l-1} \mathbf{i}_{n_l-1} \xrightarrow{\text{HMAP}^{(n_l \rightarrow n_h)}} \\
c^{(n_h)} &:= a_0 + a_1 \mathbf{i}_1 + ... + a_{n_l-1} \mathbf{i}_{n_l-1} + \mathbf{0} \mathbf{i}_{n_l} + ... + \mathbf{0} \mathbf{i}_{n_h-1}
\end{aligned}
\tag{1}
$$

### 3.1.2 HYPERCOMPLEX LINEAR LAYER

We generalize the linear layer to hypercomplex spaces of arbitrary dimension $n$. The linear transformation from $d_1$ to $d_2$ space is expressed as:

$$
\mathbf{O}^{(n)} = \mathbf{W}^{(n)^T} \mathbf{X}^{(n)} + \mathbf{b}^{(n)}
\tag{2}
$$

where the input vector $\mathbf{X}^{(n)}$, the mapping matrix $\mathbf{W}^{(n)}$, the bias matrix $\mathbf{b}^{(n)}$, and the output $\mathbf{O}^{(n)}$ are all composed of hypercomplex numbers of the same dimension $n$

$$
\begin{bmatrix} c_{o;0}^n \\ c_{o;1}^n \\ \vdots \\ c_{o;d_2}^n \end{bmatrix}
=
\begin{bmatrix} c_{w;0,0}^n, c_{w;0,1}^n, \cdots, c_{w;0,d_1}^n \\ c_{w;1,0}^n, c_{w;1,1}^n, \cdots, c_{w;1,d_1}^n \\ \vdots \\ c_{w;d_2,0}^n, c_{w;d_2,1}^n, \cdots, c_{w;d_2,d_1}^n \end{bmatrix}
\begin{bmatrix} c_{x;0}^n \\ c_{x;1}^n \\ \vdots \\ c_{x;d_1}^n \end{bmatrix}
+
\begin{bmatrix} c_{b;0}^n \\ c_{b;1}^n \\ \vdots \\ c_{b;d_2}^n \end{bmatrix}
\tag{3}
$$

It is evident that the output of the hypercomplex linear layer can still be represented in a manner analogous to real-valued matrix multiplication:

$$
c_{o;i}^{(n)} = c_{w;i,0}^{(n)} \times c_{x;0}^{(n)} + \cdots + c_{w;i,d_1}^{(n)} \times c_{x;d_1}^{(n)} + c_{b;i}^{(n)}
\tag{4}
$$

where the product of two hypercomplex numbers must conform to the multiplication rules of the hypercomplex space. In Appendix G, we present comprehensive techniques for generating multiplication rules in hypercomplex spaces of any power-of-two dimension, along with an efficient computational approach for hypercomplex linear mappings.

For simplicity, we consistently represent the linear layer in an $n$-dimensional hypercomplex space as

$$
\text{HLinear}^{(n)}(X^{(n)})
\tag{5}
$$

### 3.1.3 HYPERCOMPLEX NORM TANH ACTIVATION FUNCTION

To capture non-linear mapping relationships in hypercomplex spaces, we introduce the Hypercomplex Norm Tanh (HNTanh) activation function. This function applies non-linear transformations to both

the real and multiple imaginary components of the hypercomplex number. The computation process of HNTanh$^{(n)}$ is defined as:

$$\hat{a}_i = \frac{a_i}{\|c^{(n)}\|_p} \tanh(\|c^{(n)}\|_p) \tag{6}$$

where $a_i$ denotes the coefficients of the hypercomplex number, and $\| \cdot \|_p$ represents the $p$-norm. HNTanh enables non-linear transformation of the modulus of the hypercomplex number while maintaining its phase. When the hypercomplex number simplifies to the real number space, this activation function becomes equivalent to the $\texttt{tanh}$ activation function. Additionally, this function is both continuous and real-differentiable. Relevant theoretical derivations and advantage analyses are provided in the Appendix I and H.1.

### 3.1.4 LOW-DIMENSIONAL HYPERCOMPLEX MAPPING

Similar to the process of mapping to higher dimensions, when mapping to low-dimensional hypercomplex spaces, we follow the principle of retaining the lower-dimensional coefficients while discarding the higher-dimensional ones. This operation is denoted as LMAP$^{(n_h \to n_l)}$.

$$c^{(n_h)} := a_0 + a_1\mathbf{i}_1 + ... + a_{n_l-1}\mathbf{i}_{n_l-1} + a_{n_l}\mathbf{i}_{n_l} + ... + a_{n_h-1}\mathbf{i}_{n_h-1}$$
$$\xrightarrow{\text{LMAP}^{(n_h \to n_l)}} c^{(n_l)} := a_0 + a_1\mathbf{i}_1 + ... + a_{n_l-1}\mathbf{i}_{n_l-1} \tag{7}$$

where $n_h$ and $n_l$ represent the dimensions of the hypercomplex numbers, and $n_h > n_l$.

### 3.1.5 REAL-HYPERCOMPLEX-REAL DOMAIN MULTI-LAYER PERCEPTRON

The Real-Hypercomplex-Real domain Multi-Layer Perceptron (RHR-MLP) architecture we utilize comprises High-Dimensional Hypercomplex Mapping, $L\times$ Hypercomplex Linear Layers with Hypercomplex Norm Tanh Activation Function and Dropout, a prediction layer, and Low-Dimensional Hypercomplex Mapping, as depicted in Figure 2. We denote the RHR-MLP in the $n$-dimensional hypercomplex space as RHR-MLP$^{(n)}(\mathbf{X})$, and its computational process is as follows:

$$\begin{aligned}
\mathbf{X}_{(0)}^{(n)} &= \text{HMAP}^{(1 \to n)}(\mathbf{X}) \\
\mathbf{X}_{(i)}^{(n)} &= \text{Dropout}(\text{HNTanh}^{(n)}(\text{HLinear}^{(n)}(\mathbf{X}_{(i-1)}^{(n)}))) \quad i = 1, 2, ..., L \\
\mathbf{O}^{(n)} &= \text{HLinear}^{(n)}([\, \mathbf{X}_{(1)}^{(n)} | \cdots | \mathbf{X}_{(L)}^{(n)} \,]) \\
\mathbf{O} &= \text{LMAP}^{(n \to 1)}(\mathbf{O}^{(n)})
\end{aligned} \tag{8}$$

where $[\cdot|\cdot]$ represents matrix concatenation.

## 3.2 NUMERION ARCHITECTURE

We introduce a multi-hypercomplex space time series prediction model, named **Numerion**, which transcends the constraints of the real number domain by mapping sequences to complex, quaternion, and higher-dimensional hypercomplex spaces. This approach seeks to uncover distinct latent temporal patterns inherent in various hypercomplex spaces and adaptively integrate multi-dimensional features. The model architecture, as depicted in Figure 2, primarily comprises three components: Multi-Level Patch Embedding, Multi-dimensional RHR-MLP, and Multi-Hypercomplex Adaptive Fusion. Prior to inputting the sequence and after generating the prediction, we apply mean normalization and denormalization, respectively.

### 3.2.1 MULTI-LEVEL PATCH EMBEDDING

Time series often exhibit both long-term and short-term characteristics, which are challenging for a simple MLP structure to model directly. However, PatchNie et al. (2022) enables segmentation of the time series into varying lengths, allowing the MLP to capture temporal features across different scales. Thus, before mapping the time series to multi-hypercomplex spaces, we enhance long-term and short-term features through Multi-Level Patch EmbeddingNie et al. (2022).

We denote the number of patch levels as $l_p$ and define the segmentation length for each level as $\lfloor \frac{T}{2^i} \rfloor, i = 0, 1, ..., l_p - 1$. The time series is segmented according to the length of each level, forming multiple subsequences, where sequences within the same level have identical lengths. We linearly map the time series at each level to a unified encoding dimension $d_e$, average the encodings of multiple sequences at the same level, and concatenate the encodings from different levels to form a time series $\mathbf{X}_p \in \mathbb{R}^{F \times (d_e \cdot l_p)}$ that incorporates features of varying periodicities.

### 3.2.2 MULTI-DIMENSIONAL RHR-MLP

Next, $\mathbf{X}_p$ is fed into the multi-dimensional RHR-MLP to uncover latent patterns in hypercomplex spaces of varying dimensions. Given the stepwise derivation of hypercomplex spaces, we select consecutive power-of-two spaces starting from real numbers: Real Numbers (Real), Complex Numbers (Comp), Quaternions (Quat), Octonions (Octo), and Sedenions (Sede).

$$\mathbf{O}_{Real} = \text{RHR-MLP}^{(1)}(\mathbf{X}_p), \; \mathbf{O}_{Comp} = \text{RHR-MLP}^{(2)}(\mathbf{X}_p), \; \mathbf{O}_{Quat} = \text{RHR-MLP}^{(4)}(\mathbf{X}_p)$$
$$\mathbf{O}_{Octo} = \text{RHR-MLP}^{(8)}(\mathbf{X}_p), \; \mathbf{O}_{Sede} = \text{RHR-MLP}^{(16)}(\mathbf{X}_p) \tag{9}$$

where $\mathbf{O}_{Real}, \mathbf{O}_{Comp}, \mathbf{O}_{Quat}, \mathbf{O}_{Octo}$, and $\mathbf{O}_{Sede}$ denote the prediction outputs in each hypercomplex space. All RHR-MLPs operate in parallel, independently uncovering latent temporal patterns in their respective spaces. We avoid higher-dimensional spaces for two reasons: first, computational time and memory demands escalate rapidly; second, experiments reveal that higher-dimensional spaces tend to model lower-frequency features, as shown in Section 4.2, and the benefits of excessively low-frequency features for time series prediction are limited.

### 3.2.3 MULTI-HYPERCOMPLEX ADAPTIVE FUSION AND LOSS

We design an adaptive fusion mechanism to integrate features from different spaces and generate the final prediction. The outputs from all spaces are stacked, and an MLP learns adaptive weights for each space. The softmax function ensures these weights sum to 1, and the weighted sum of the outputs yields the fused prediction result.

$$\mathbf{O}_s = <\mathbf{O}_{Real}, \mathbf{O}_{Comp}\mathbf{O}_{Quat}, \mathbf{O}_{Octo}, \mathbf{O}_{Sede}>$$
$$\mathbf{R} = \mu(\mathbf{W}_{f,2}^T \; \sigma(\mathbf{W}_{f,1}^T \mathbf{O}_s + \mathbf{b}_{f,1}) + \mathbf{b}_{f,2}), \; \hat{\mathbf{Y}} = \text{sum}(\mathbf{R} * \mathbf{O}_s) \tag{10}$$

where $< \cdot >$ denotes matrix stacking, $\text{sum}(\cdot)$ represents the summation operation, $\mu(\cdot)$ is the softmax activation function, $\sigma(\cdot)$ is the GeLU activation function, and $\hat{\mathbf{Y}}$ is the model's prediction result. The model is trained using the MAE Loss as the loss function.

Regarding computational complexity, Numerion operates with an asymptotic time complexity of $\mathcal{O}(L \times d)$, where $L$ represents the time series length and $d$ denotes the encoding dimension. While hypercomplex multiplication inherently scales quadratically with the chosen algebra dimension $n$, this dimension acts as a fixed structural constant rather than a variable dependent on input length. Consequently, the $\mathcal{O}(n^2)$ cost introduces only a constant multiplier that does not alter the model's fundamental linear scaling $\mathcal{O}(L)$ with respect to the sequence length. This preserves a crucial asymptotic advantage over the $\mathcal{O}(L^2)$ scaling typical of standard Transformer architectures. Furthermore, because these hypercomplex operations are formulated as highly parallelizable matrix multiplications, modern GPU infrastructure effectively absorbs this constant computational overhead during training.

## 4 EXPERIMENT

In Section 4.1, we present the comparisons with other methods. Section 4.2 reveals the multi-frequency natural decomposition patterns of time series through visualization and quantitative analysis. Section 4.3 and Section 4.4 focus on parameter experiments and ablation studies, respectively. Section 4.5 analyzes efficiency. In Appendix G, we introduce a High-Efficient Hypercomplex Linear Layers approach to accelerate hypercomplex computations. Appendix H provides theoretical supplements. Appendix I includes a detailed analysis of HNTanh. Appendix J offers additional visualizations to uncover the natural decomposition patterns of time series. Appendix K provides quantitative statistics across various hypercomplex spaces. Appendix F reports the error bars of our model results.

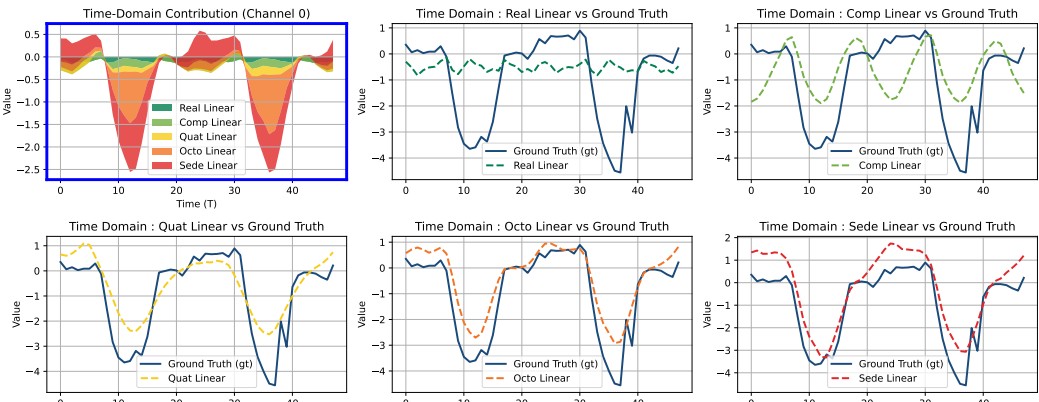

Figure 3: Visualization of MLP outputs on ETTh1 (channel 0). The first plot shows the weighted sum of five RHR-MLPs, with color indicating the module and thickness representing its contribution

## 4.1 MAIN RESULT

Table 1: Results of the Long-Term Time Series Forecasting task. We averaged the results across four prediction lengths: {96, 192, 336, 720}. The best result is indicated in **bold**, and the second-best result is underlined.

| | Numerion | | PatchMLP | | TimeMixer | | iTransformer | | TexFilter | | PaiFilter | | PatchTST | | Crossformer | | TimesNet | | FEDformer | | DLinear | |
|---|---|---|---|---|---|---|---|---|---|---|---|---|---|---|---|---|---|---|---|---|---|
| | MAE | MSE | MAE | MSE | MAE | MSE | MAE | MSE | MAE | MSE | MAE | MSE | MAE | MSE | MAE | MSE | MAE | MSE | MAE | MSE | MAE | MSE |
| ETTh1 | **0.417** | **0.414** | 0.437 | 0.456 | 0.440 | 0.447 | 0.467 | 0.454 | 0.439 | 0.441 | 0.432 | 0.440 | 0.454 | 0.469 | 0.522 | 0.529 | 0.450 | 0.458 | 0.484 | 0.498 | 0.452 | 0.456 |
| ETTh2 | **0.388** | **0.364** | 0.397 | 0.375 | 0.409 | 0.388 | 0.407 | 0.383 | 0.407 | 0.383 | 0.404 | 0.378 | 0.407 | 0.387 | 0.684 | 0.942 | 0.427 | 0.414 | 0.449 | 0.437 | 0.515 | 0.559 |
| ETTm1 | **0.378** | **0.370** | 0.391 | 0.394 | 0.396 | 0.381 | 0.410 | 0.410 | 0.401 | 0.392 | 0.398 | 0.384 | 0.400 | 0.387 | 0.495 | 0.513 | 0.406 | 0.400 | 0.452 | 0.448 | 0.407 | 0.403 |
| ETTm2 | **0.315** | **0.270** | 0.326 | 0.282 | 0.323 | 0.275 | 0.332 | 0.288 | 0.328 | 0.285 | 0.322 | 0.276 | 0.326 | 0.281 | 0.611 | 0.757 | 0.333 | 0.291 | 0.349 | 0.305 | 0.401 | 0.350 |
| Weather | **0.271** | 0.246 | 0.272 | 0.249 | 0.277 | 0.246 | 0.278 | 0.258 | 0.272 | **0.245** | 0.274 | 0.248 | 0.281 | 0.259 | 0.320 | 0.264 | 0.284 | 0.259 | 0.360 | 0.309 | 0.317 | 0.265 |
| Solar Energy | **0.262** | 0.252 | 0.292 | 0.309 | 0.278 | **0.242** | 0.286 | 0.295 | 0.284 | 0.295 | 0.286 | 0.298 | 0.290 | 0.277 | 0.307 | 0.270 | 0.442 | 0.406 | 0.374 | 0.403 | 0.383 | 0.328 | 0.401 | 0.330 |
| Electricity | **0.267** | 0.181 | 0.275 | 0.190 | 0.273 | 0.182 | 0.270 | **0.178** | 0.285 | 0.198 | 0.278 | 0.197 | 0.290 | 0.205 | 0.334 | 0.244 | 0.304 | 0.193 | 0.327 | 0.214 | 0.300 | 0.212 |
| Traffic | **0.281** | 0.468 | 0.298 | 0.485 | 0.298 | 0.485 | 0.282 | **0.428** | 0.300 | 0.470 | 0.340 | 0.521 | 0.304 | 0.481 | 0.426 | 0.667 | 0.336 | 0.620 | 0.376 | 0.610 | 0.383 | 0.625 |
| Exchange | **0.399** | 0.358 | 0.432 | 0.416 | 0.453 | 0.391 | 0.470 | 0.519 | 0.464 | 0.515 | 0.414 | 0.379 | 0.404 | 0.367 | 0.707 | 0.940 | 0.443 | 0.416 | 0.429 | 0.519 | 0.414 | **0.354** |

Table 1 showcases the comparison results of our method against other approaches, with all outcomes representing the average across four prediction lengths {96, 192, 336, 720}. The complete experimental results are detailed in Appendix B and error bar are provided in Appendix F. It is evident that our method achieves the best MAE across all datasets, particularly on datasets such as ETTh1 and ETTh2, where the performance improvement is substantial, with an overall average MAE enhancement exceeding 4%. Our method consistently ranks within the top two in terms of MSE, demonstrating an improvement of over 3% on the ETT datasets. On datasets with more variables, such as Traffic, our method, being variable-independent, occasionally exhibits slightly lower MSE performance compared to variable-fusion models like timemixer and itransformer. Nonetheless, our method demonstrates significant advancements over other models.

## 4.2 VISUALIZATION ANALYSIS OF RHR-MLP

To better understand the functional roles of different RHR-MLP in our model, we visualized the outputs of each RHR-MLP under the best-performing configuration. As shown in Figure 3 and detailed further in Appendix J.1, a clear frequency-based specialization emerges: high-dimensional hypercomplex layers are primarily responsible for capturing low-frequency, low-dimensional trend components, whereas real and complex-valued layers focus on high-frequency, high-dimensional fluctuations, such as local seasonal signals and noise. This emergent decomposition enables the model to preserve long-term trend accuracy without compromising the fit on short-term patterns.

Quantitatively, the visualization shows that hypercomplex layers contribute about 60% of the final output, with the hexadecimal layer alone accounting for nearly 40%, underscoring its dominant role in modeling temporal trends. In contrast, the real and complex layers—specialized for mid- to high-frequency content—contribute the remaining 40%. This separation is not due to any handcrafted constraints or predefined modules. Instead, it emerges naturally from training, allowing the model to

learn a flexible statistical decomposition directly from data. We offer theoretical insights into this phenomenon in Appendix L.1, conjecturing that hypercomplex representations naturally introduce an implicit stable rank reduction and low-pass bias via symmetry-induced spectral degeneracies. This structural perspective helps explain the method's built-in regularization, robustness to measurement noise, and strong capacity for modeling long-horizon trends, standing in contrast to classical methods that rely on rigid assumptions. To further support these visualization results, we also present a quantitative statistical analysis in Appendix K, which independently reaches a similar conclusion, reinforcing our interpretation.

These insights suggest that low-frequency components may inherently pose greater modeling challenges, and that addressing limitations in low-frequency prediction could further enhance the model's overall capacity—including its ability to represent high-frequency structures. Future work may refine the allocation of representational resources to better balance the frequency spectrum.

## 4.3 HYPER-PARAMETER STUDY

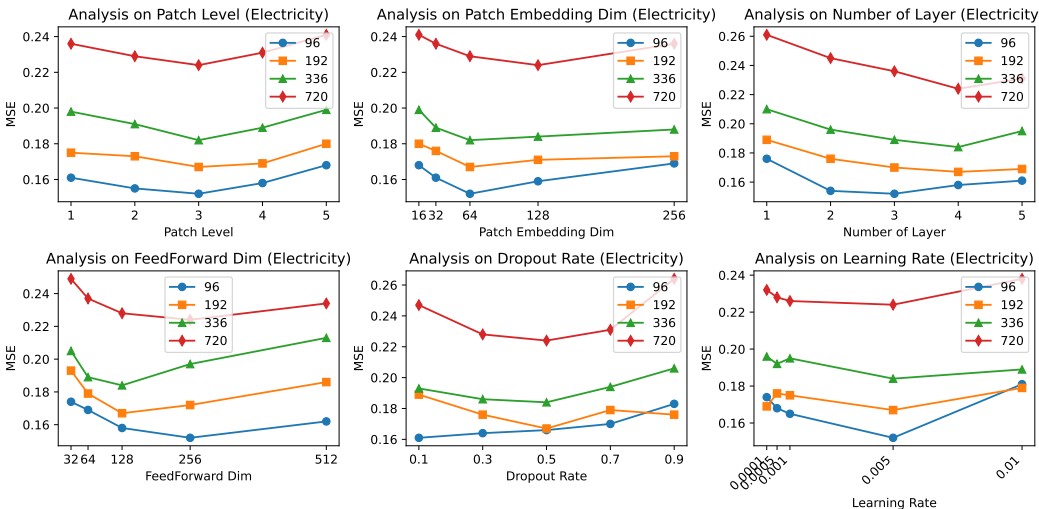

Figure 4: Hyper-parameter Study on Electricity with P = {96, 192, 336, 720}

We performed a sensitivity analysis on the Electricity dataset for four prediction lengths, focusing on key parameters: patch level $l_p$ in Multi-Level Patch Embedding, patch embedding dimension $d_e$, RHR-MLP layer number, feedforward dimension, dropout rate, and learning rate. The results, shown in Figure 4, indicate that the Numerion model is less sensitive to patch-related parameters, favoring larger dimensions to capture more information, though performance gains are marginal. Conversely, it is more sensitive to RHR-MLP parameters. Additionally, Numerion benefits from larger dropout rates and lower learning rates, improving its ability to fit and generalize in high-dimensional spaces. Additional parameter studys are provided in Appendix C.

## 4.4 ABLATION STUDY

We conducted ablation experiments on ETTh1 (P=720) and Electricity (P=96) to evaluate each component of Numerion (Table 2). Removing the Multi-Level Patch Embedding caused sharp degradation, confirming its role in capturing multi-scale temporal cues. Excluding any power-of-two RHR-MLP also weakened performance, showing that distinct hypercomplex spaces specialize in different temporal patterns, with higher dimensions particularly effective for low-frequency periodicity. Adaptive Fusion proved essential, as its removal greatly harmed results on Electricity. A detailed breakdown of 17 ablation cases is in Appendix D.

We also ablated the hypercomplex settings (Table 6, see Appendix D). Removing the HNTanh activation—or either of its parts, tanh on the modulus or $p$-norm normalization—consistently degraded performance, despite minor dataset-specific gains (e.g., ETTm1). Full HNTanh provided the most

Table 2: Ablation Study

| | ETTh1 with P=720 | | | | Electricity with P=96 | | | |
| --- | --- | --- | --- | --- | --- | --- | --- | --- |
| | MSE | | MAE | | MSE | | MAE | |
| **Numerion** | **0.449** | | **0.449** | | **0.152** | | **0.239** | |
| w/o Multi-Level Patch | 0.455 | -1.34% | 0.456 | -1.56% | 0.164 | -7.89% | 0.250 | -4.60% |
| w/o Real | 0.451 | -0.45% | 0.451 | -0.45% | 0.160 | -5.26% | 0.246 | -2.93% |
| w/o Comp | 0.451 | -0.45% | 0.452 | -0.67% | 0.156 | -2.63% | 0.244 | -2.09% |
| w/o Quat | 0.453 | -0.89% | 0.451 | -0.45% | 0.160 | -5.26% | 0.246 | -2.93% |
| w/o Octo | 0.453 | -0.89% | 0.455 | -1.34% | 0.161 | -5.92% | 0.247 | -3.35% |
| w/o Sede | 0.454 | -1.11% | 0.456 | -1.56% | 0.166 | -9.21% | 0.252 | -5.44% |
| w/o Adaptive Fusion | 0.450 | -0.22% | 0.451 | -0.45% | 0.177 | -16.45% | 0.263 | -10.04% |

stable improvements, underscoring the synergy of both components. To isolate dimensional effects, we expanded real inputs for a standard MLP with comparable parameters. Its inferior results confirmed that hypercomplex gains stem not from dimensionality but from structured multiplication and phase coupling, which enforce stronger inductive bias for frequency decomposition and multichannel interactions.

## 4.5 EFFICIENCY ANALYSIS

We assessed the efficiency of our model in comparison with several baseline models using a single NVIDIA A5000 GPU and Intel Xeon Gold 6326 CPU. As illustrated in Figure 5, our model exhibits competitive performance while maintaining reasonable resource consumption. Although it incurs higher computational overhead and parameter count due to the simulation of hypercomplex operations using real numbers, its overall efficiency remains practical and acceptable for real-world applications. It is important to contextualize this overhead: the inflation in parameters and memory is primarily a software-level artifact stemming from the current lack of native framework support and hardware acceleration for multi-dimensional hypercomplex algebra. Despite the necessity of simulating these spaces relative to standard real-valued linear layers, Numerion's MLP-based architecture inherently preserves a highly favorable computational complexity compared to heavier Transformer- or Mamba-based models. Furthermore, these resource constraints are not fundamental architectural limitations. As the field increasingly recognizes the representational power of hypercomplex spaces, future dedicated low-level implementations will significantly reduce memory and time requirements, mirroring the historical optimization trajectory of complex-valued neural networks. Detailed reasoning behind these design choices and efficiency trade-offs is provided in the Appendix E.

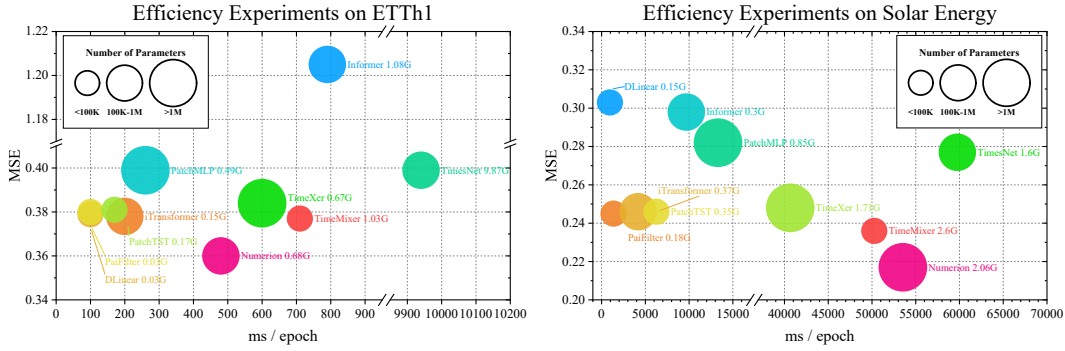

Figure 5: Efficiency Analysis on ETTh1 and Solar Energy dataset

## 5 RELATED WORK

### 5.1 TIME SERIES FORECASTING

In this section, we classify most time series forecasting tasks based on whether they adopt the standard benchmark datasets widely recognized by the academic community, as used in our work. We focus particularly on those studies that inspired the design of our model. First, a number of classical models

based on MLP architectures are closely related to ours. These include N-BEATSOreshkin et al. (2019), N-HITSChallu et al. (2023), MLinearLi et al. (2023), DLinearZeng et al. (2023), PatchMLPTang & Zhang (2025), FilterNetYi et al. (2024), TimesNetWu et al. (2022), TimeMixerWang et al. (2024c), and its extension TimeMixer++Wang et al. (2024b). Among them, TimeMixer combines linear and sampling layers to implement bottom-up and top-down mixing, achieving a unique trend-seasonality decomposition that guided our exploration. MLinear adopts a Mixture of Experts (MoE) framework that combines predictions from multiple linear layers. Its design and aggregation mechanism provided key insights into the architectural design of our own model.

Secondly, Transformer-based models form another major line of research in time series forecasting. Prominent models in this category include InformerZhou et al. (2021), AutoformerWu et al. (2021), FEDformerZhou et al. (2022b), CrossFormerZhang & Yan (2023) , PatchTSTNie et al. (2022), STAEformerLiu et al. (2023a), iTransformerLiu et al. (2023b), TimeXerWang et al. (2024d), and Timer-XLLiu et al. (2024). Notable examples such as AutoformerWu et al. (2021), FEDformerZhou et al. (2022b), and PatchTSTNie et al. (2022) explored trend-season decomposition and hierarchical embeddings tailored for time series, laying groundwork that we build upon. Other models like iTransformerLiu et al. (2023b) and Timer-XLLiu et al. (2024) address temporal ordering in multivariate settings, providing insights into sequence processing strategies that inform our design. Many of the latest methods, such as FBMYang et al. (2024), have mined undiscovered temporal patterns from the frequency domain.

Several recent works based on the novel Mamba architecture, such as TimeMachineAhamed & Cheng (2024), HIGSTMYan et al. (2025b) and Bi-MambaTang et al. (2024), as well as those using spatio-temporal graph convolutional networks like AGCRNBai et al. (2020) and ASTGCNGuo et al. (2019), have contributed meaningfully to the field. Other advances include multi-channel aggregation methods (e.g., CCMChen et al. (2024)), contrastive learning techniques (RCLYan et al. (2025a)), and label refinement approaches (FreDFWang et al. (2024a)). These works have all made outstanding contributions to time series forecasting.

## 5.2 Hypercomplex Neural Networks

Although hypercomplex neural networks are not yet widely adopted in machine learning—partly due to the challenges discussed in Appendix I—they offer valuable insights relevant to our work. One recent study Zhang et al. (2021) proposes a parameter-efficient design with learnable hypercomplex number multiplication, inspiring improvements in our model structure. Another work Lopez et al. (2024) introduces a cosine similarity transformation aligned with feature weights, enabling direct visualization of attention in quaternion convolutional layers and guiding our interpretability approach. Additional research on hypercomplex networks Comminiello et al. (2024), including quaternion convolutions Altamirano-Gomez & Gershenson (2023), quaternion transformers Singh et al. (2024), and octonion and sedenion networks Popa (2016); Pavlov et al. (2023), also contributes ideas, particularly for our activation function design. A more detailed theoretical connection between hypercomplex representations and time series modeling is provided in Appendix L.2.

## 6 Conclusion

In this paper, we extend linear layers and the Tanh activation to hypercomplex spaces of arbitrary power-of-two dimensions, providing a unified framework with the real domain and an efficient computation method. Building on this, we propose the Real-Hypercomplex-Real Multi-Layer Perceptron (RHR-MLP), which transforms inputs into hypercomplex spaces for enhanced modeling. This leads to Numerion, a forecasting model that integrates multiple hypercomplex spaces to decompose, model, and adaptively fuse multi-frequency components. Numerion is simple and interpretable, with experiments confirming its effectiveness and showing that higher-dimensional spaces capture low-frequency features. Looking forward, as noted in Appendix N, we expect progress from exploring the characteristics of hypercomplex spaces, advancing theoretical analysis of their representational power, and developing optimizers and hardware tailored to hypercomplex arithmetic.

ACKNOWLEDGMENTS

This work is supported by the National Natural Science Foundation of China (Grant No. 62276008, 62250037, and 62076010), and partially supported by the National Key R&D of China (Grant #2022YFF0800601).

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

APPENDIX

## A EXPERIMENTS SETTING SUPPLEMENTS

### A.1 DATASETS

We evaluated the performance of the model on nine publicly available long-term time series forecasting datasets: Weather, Solar Energy, Electricity, Traffic, Exchange, ETTh1, ETTh2, ETTm1, and ETTm2. Detailed information about the datasets is provided in Table 3.

Table 3: Basic Information of Dataset.

| Dataset | Data Partition | Frequency | Feature Dim | Type |
|---|---|---|---|---|
| Weather | (36792,5271,10540) | 10min | 21 | Weather |
| Solar Energy | (36601,5161,10417) | 10min | 137 | Electricity |
| Electricity | (18317,2633,5261) | Hourly | 321 | Electricity |
| Traffic | (12185,1757,3509) | Hourly | 862 | Traffic |
| Exchange | (5120,665,1422) | Daily | 8 | Weather |
| ETTh1 | (8545,2881,2881) | 15min | 7 | Temperature |
| ETTh2 | (8545,2881,2881) | 15min | 7 | Temperature |
| ETTm1 | (34465,11521,11521) | 15min | 7 | Temperature |
| ETTm2 | (34465,11521,11521) | 15min | 7 | Temperature |

### A.2 BASELINES

We selected a variety of state-of-the-art baselines to compare with our method in order to evaluate the performance of Numerion. Specifically, we utilize Transformer-based models: FEDformer, Crossformer, iTransformer, and PatchTST; convolutional neural network (CNN)-based models: TimesNet; MLP-based models: TimeMixer, PatchMLP, and DLinear; as well as filter-based models: TexFilter and PaiFilter.

### A.3 METRICS

We evaluated all models using Mean Squared Error (MSE) and Mean Absolute Error (MAE) as the evaluation metrics.

$$\mathbf{MAE} = \frac{1}{N} \sum_{i=1}^{N} |\mathbf{Y}_i - \hat{\mathbf{Y}}_i|$$

$$\mathbf{MSE} = \frac{1}{N} \sum_{i=1}^{N} \left(\mathbf{Y}_i - \hat{\mathbf{Y}}_i\right)^2$$

### A.4 EXPERIMENT DETAILS

We repeat each experiment three times on a server equipped with an NVIDIA GeForce RTX 3090 GPU and an AMD EPYC 7282 16-Core Processor. We employ Adam as the optimizer, with the learning rate set between 1e-4 and 1e-2 depending on the dataset. For Numerion, the patch level is configured between 1 and 4, the patch embedding dimension is set between 64 and 256, the number of RHR-MLP layers is set to 2 or 3, the initial dimension of RHR-MLP is set between 64 and 256 with subsequent HLinear dimensions decreasing, and the dropout rate is set between 0.5 and 0.7. For datasets with a feature number greater than 100, the batch size is set to 100, while for other datasets, the batch size is set to 512. The HLinear layers are initialized using random sampling from a standard normal distribution, followed by normalization, and the biases are initialized as zero vectors.

## B FULL RESULTS

Table 4 provides the detailed comparison results on the Long-Term Time Series Forecasting task. We set the input length to 96 and the prediction lengths to {96, 192, 336, 720}. Avg is averaged from all four prediction lengths.

Table 4: Full results for the Long-Term Time Series Forecasting task. We compare competitive models under different prediction lengths :{96,192,336,720}. Avg is averaged from all four prediction lengths.

| Model | P | Numerion | | PatchMLP | | TimeMixer | | iTransformer | | TexFilter | | PaiFilter | | PatchTST | | Crossformer | | TimesNet | | FEDformer | | DLinear | |
|---|---|---|---|---|---|---|---|---|---|---|---|---|---|---|---|---|---|---|---|---|---|---|---|
| | | MAE | MSE | MAE | MSE | MAE | MSE | MAE | MSE | MAE | MSE | MAE | MSE | MAE | MSE | MAE | MSE | MAE | MSE | MAE | MSE | MAE | MSE |
| ETTh1 | 96 | 0.380 | 0.359 | 0.390 | 0.385 | 0.400 | 0.375 | 0.405 | 0.386 | 0.402 | 0.382 | 0.394 | 0.375 | 0.419 | 0.414 | 0.448 | 0.423 | 0.402 | 0.384 | 0.424 | 0.395 | 0.400 | 0.386 |
| | 192 | 0.409 | 0.407 | 0.431 | 0.458 | 0.421 | 0.429 | 0.512 | 0.441 | 0.429 | 0.430 | 0.422 | 0.436 | 0.445 | 0.460 | 0.474 | 0.471 | 0.429 | 0.436 | 0.470 | 0.469 | 0.432 | 0.437 |
| | 336 | 0.429 | 0.444 | 0.460 | 0.500 | 0.458 | 0.484 | 0.458 | 0.487 | 0.451 | 0.472 | 0.443 | 0.476 | 0.466 | 0.501 | 0.546 | 0.570 | 0.469 | 0.491 | 0.499 | 0.530 | 0.459 | 0.481 |
| | 720 | 0.449 | 0.447 | 0.468 | 0.482 | 0.482 | 0.498 | 0.491 | 0.503 | 0.473 | 0.481 | 0.469 | 0.474 | 0.488 | 0.500 | 0.621 | 0.653 | 0.500 | 0.521 | 0.544 | 0.598 | 0.516 | 0.519 |
| | avg | 0.417 | 0.414 | 0.437 | 0.456 | 0.440 | 0.447 | 0.467 | 0.454 | 0.439 | 0.441 | 0.432 | 0.440 | 0.454 | 0.469 | 0.522 | 0.529 | 0.450 | 0.458 | 0.484 | 0.498 | 0.452 | 0.456 |
| ETTh2 | 96 | 0.326 | 0.279 | 0.344 | 0.296 | 0.348 | 0.296 | 0.349 | 0.297 | 0.343 | 0.293 | 0.343 | 0.292 | 0.348 | 0.302 | 0.584 | 0.745 | 0.374 | 0.340 | 0.397 | 0.358 | 0.387 | 0.333 |
| | 192 | 0.379 | 0.359 | 0.390 | 0.371 | 0.391 | 0.371 | 0.400 | 0.380 | 0.396 | 0.374 | 0.395 | 0.369 | 0.400 | 0.388 | 0.656 | 0.877 | 0.414 | 0.402 | 0.439 | 0.429 | 0.476 | 0.477 |
| | 336 | 0.416 | 0.406 | 0.424 | 0.417 | 0.428 | 0.417 | 0.432 | 0.428 | 0.430 | 0.417 | 0.432 | 0.420 | 0.433 | 0.426 | 0.731 | 1.043 | 0.452 | 0.452 | 0.487 | 0.496 | 0.541 | 0.594 |
| | 720 | 0.433 | 0.414 | 0.432 | 0.415 | 0.468 | 0.469 | 0.445 | 0.427 | 0.460 | 0.449 | 0.446 | 0.430 | 0.446 | 0.431 | 0.763 | 1.104 | 0.468 | 0.462 | 0.474 | 0.463 | 0.657 | 0.831 |
| | avg | 0.388 | 0.364 | 0.397 | 0.375 | 0.409 | 0.388 | 0.407 | 0.383 | 0.407 | 0.383 | 0.404 | 0.378 | 0.407 | 0.387 | 0.684 | 0.942 | 0.427 | 0.414 | 0.449 | 0.437 | 0.515 | 0.559 |
| ETTm1 | 96 | 0.337 | 0.305 | 0.348 | 0.316 | 0.357 | 0.320 | 0.368 | 0.334 | 0.361 | 0.321 | 0.358 | 0.318 | 0.367 | 0.329 | 0.426 | 0.404 | 0.375 | 0.338 | 0.419 | 0.379 | 0.372 | 0.345 |
| | 192 | 0.367 | 0.356 | 0.371 | 0.363 | 0.381 | 0.361 | 0.393 | 0.390 | 0.387 | 0.367 | 0.383 | 0.364 | 0.385 | 0.367 | 0.451 | 0.450 | 0.387 | 0.374 | 0.441 | 0.426 | 0.389 | 0.380 |
| | 336 | 0.386 | 0.380 | 0.397 | 0.397 | 0.404 | 0.390 | 0.420 | 0.426 | 0.409 | 0.401 | 0.406 | 0.396 | 0.410 | 0.399 | 0.515 | 0.532 | 0.411 | 0.410 | 0.459 | 0.445 | 0.413 | 0.413 |
| | 720 | 0.423 | 0.439 | 0.448 | 0.499 | 0.441 | 0.454 | 0.459 | 0.491 | 0.448 | 0.477 | 0.444 | 0.456 | 0.439 | 0.454 | 0.589 | 0.666 | 0.450 | 0.478 | 0.490 | 0.543 | 0.453 | 0.474 |
| | avg | 0.378 | 0.370 | 0.391 | 0.394 | 0.396 | 0.381 | 0.410 | 0.410 | 0.401 | 0.392 | 0.398 | 0.384 | 0.400 | 0.387 | 0.495 | 0.513 | 0.406 | 0.400 | 0.452 | 0.448 | 0.407 | 0.403 |
| ETTm2 | 96 | 0.249 | 0.170 | 0.259 | 0.178 | 0.258 | 0.175 | 0.264 | 0.180 | 0.258 | 0.175 | 0.257 | 0.174 | 0.259 | 0.175 | 0.366 | 0.287 | 0.267 | 0.187 | 0.287 | 0.203 | 0.292 | 0.193 |
| | 192 | 0.292 | 0.231 | 0.304 | 0.242 | 0.299 | 0.237 | 0.309 | 0.250 | 0.301 | 0.240 | 0.300 | 0.240 | 0.302 | 0.241 | 0.492 | 0.414 | 0.309 | 0.249 | 0.328 | 0.269 | 0.362 | 0.284 |
| | 336 | 0.330 | 0.289 | 0.341 | 0.302 | 0.340 | 0.298 | 0.348 | 0.311 | 0.347 | 0.311 | 0.339 | 0.297 | 0.343 | 0.305 | 0.542 | 0.597 | 0.351 | 0.321 | 0.366 | 0.325 | 0.427 | 0.369 |
| | 720 | 0.387 | 0.390 | 0.399 | 0.405 | 0.396 | 0.391 | 0.407 | 0.412 | 0.405 | 0.414 | 0.393 | 0.392 | 0.400 | 0.402 | 1.042 | 1.730 | 0.403 | 0.408 | 0.415 | 0.421 | 0.522 | 0.554 |
| | avg | 0.315 | 0.270 | 0.326 | 0.282 | 0.323 | 0.275 | 0.332 | 0.288 | 0.328 | 0.285 | 0.322 | 0.276 | 0.326 | 0.281 | 0.611 | 0.757 | 0.333 | 0.291 | 0.349 | 0.305 | 0.401 | 0.350 |
| Weather | 96 | 0.203 | 0.159 | 0.204 | 0.164 | 0.214 | 0.166 | 0.214 | 0.174 | 0.207 | 0.162 | 0.210 | 0.164 | 0.218 | 0.177 | 0.271 | 0.195 | 0.211 | 0.170 | 0.296 | 0.217 | 0.255 | 0.196 |
| | 192 | 0.250 | 0.211 | 0.250 | 0.212 | 0.253 | 0.209 | 0.254 | 0.221 | 0.250 | 0.210 | 0.252 | 0.214 | 0.259 | 0.225 | 0.277 | 0.209 | 0.259 | 0.223 | 0.336 | 0.276 | 0.296 | 0.237 |
| | 336 | 0.289 | 0.270 | 0.293 | 0.269 | 0.293 | 0.264 | 0.296 | 0.278 | 0.290 | 0.265 | 0.293 | 0.268 | 0.297 | 0.278 | 0.332 | 0.273 | 0.306 | 0.280 | 0.380 | 0.339 | 0.335 | 0.283 |
| | 720 | 0.340 | 0.345 | 0.343 | 0.350 | 0.346 | 0.345 | 0.347 | 0.358 | 0.340 | 0.342 | 0.342 | 0.344 | 0.348 | 0.354 | 0.401 | 0.379 | 0.359 | 0.365 | 0.428 | 0.403 | 0.381 | 0.345 |
| | avg | 0.271 | 0.246 | 0.272 | 0.249 | 0.277 | 0.246 | 0.278 | 0.258 | 0.272 | 0.245 | 0.274 | 0.248 | 0.281 | 0.259 | 0.320 | 0.264 | 0.284 | 0.259 | 0.360 | 0.309 | 0.317 | 0.265 |
| Solar-Energy | 96 | 0.244 | 0.209 | 0.270 | 0.269 | 0.254 | 0.236 | 0.258 | 0.250 | 0.246 | 0.278 | 0.245 | 0.254 | 0.286 | 0.234 | 0.302 | 0.232 | 0.358 | 0.373 | 0.341 | 0.286 | 0.378 | 0.290 |
| | 192 | 0.261 | 0.250 | 0.293 | 0.314 | 0.284 | 0.235 | 0.286 | 0.298 | 0.283 | 0.297 | 0.281 | 0.272 | 0.310 | 0.267 | 0.410 | 0.371 | 0.376 | 0.397 | 0.337 | 0.291 | 0.398 | 0.320 |
| | 336 | 0.270 | 0.273 | 0.304 | 0.325 | 0.287 | 0.244 | 0.296 | 0.314 | 0.305 | 0.307 | 0.315 | 0.290 | 0.315 | 0.290 | 0.515 | 0.495 | 0.380 | 0.420 | 0.416 | 0.354 | 0.415 | 0.353 |
| | 720 | 0.272 | 0.274 | 0.303 | 0.327 | 0.286 | 0.253 | 0.294 | 0.319 | 0.311 | 0.309 | 0.318 | 0.292 | 0.317 | 0.289 | 0.542 | 0.526 | 0.381 | 0.420 | 0.437 | 0.380 | 0.413 | 0.356 |
| | avg | 0.262 | 0.252 | 0.292 | 0.309 | 0.278 | 0.242 | 0.284 | 0.295 | 0.286 | 0.298 | 0.290 | 0.277 | 0.307 | 0.270 | 0.442 | 0.406 | 0.374 | 0.403 | 0.383 | 0.328 | 0.401 | 0.330 |
| Electricity | 96 | 0.239 | 0.152 | 0.253 | 0.165 | 0.247 | 0.153 | 0.240 | 0.148 | 0.259 | 0.169 | 0.256 | 0.175 | 0.270 | 0.181 | 0.314 | 0.219 | 0.272 | 0.168 | 0.308 | 0.193 | 0.282 | 0.197 |
| | 192 | 0.251 | 0.167 | 0.255 | 0.168 | 0.256 | 0.166 | 0.253 | 0.162 | 0.273 | 0.183 | 0.264 | 0.182 | 0.274 | 0.188 | 0.322 | 0.231 | 0.322 | 0.184 | 0.315 | 0.201 | 0.285 | 0.196 |
| | 336 | 0.270 | 0.182 | 0.276 | 0.189 | 0.277 | 0.185 | 0.269 | 0.178 | 0.287 | 0.200 | 0.279 | 0.196 | 0.293 | 0.204 | 0.337 | 0.246 | 0.300 | 0.198 | 0.329 | 0.214 | 0.301 | 0.209 |
| | 720 | 0.308 | 0.224 | 0.315 | 0.237 | 0.310 | 0.225 | 0.317 | 0.225 | 0.322 | 0.241 | 0.313 | 0.237 | 0.324 | 0.246 | 0.363 | 0.280 | 0.320 | 0.220 | 0.355 | 0.246 | 0.333 | 0.245 |
| | avg | 0.267 | 0.181 | 0.275 | 0.190 | 0.273 | 0.182 | 0.270 | 0.178 | 0.285 | 0.198 | 0.278 | 0.197 | 0.290 | 0.205 | 0.334 | 0.244 | 0.304 | 0.193 | 0.327 | 0.214 | 0.300 | 0.212 |
| Traffic | 96 | 0.264 | 0.441 | 0.285 | 0.459 | 0.285 | 0.462 | 0.268 | 0.395 | 0.284 | 0.439 | 0.336 | 0.506 | 0.295 | 0.462 | 0.429 | 0.644 | 0.321 | 0.593 | 0.366 | 0.587 | 0.396 | 0.650 |
| | 192 | 0.276 | 0.457 | 0.287 | 0.464 | 0.296 | 0.473 | 0.276 | 0.417 | 0.293 | 0.455 | 0.333 | 0.508 | 0.296 | 0.466 | 0.431 | 0.665 | 0.336 | 0.617 | 0.373 | 0.604 | 0.370 | 0.598 |
| | 336 | 0.282 | 0.465 | 0.294 | 0.472 | 0.296 | 0.498 | 0.283 | 0.433 | 0.300 | 0.473 | 0.335 | 0.518 | 0.304 | 0.482 | 0.420 | 0.674 | 0.336 | 0.629 | 0.383 | 0.621 | 0.373 | 0.605 |
| | 720 | 0.301 | 0.507 | 0.327 | 0.547 | 0.313 | 0.506 | 0.302 | 0.467 | 0.324 | 0.512 | 0.354 | 0.553 | 0.322 | 0.514 | 0.424 | 0.683 | 0.350 | 0.640 | 0.382 | 0.626 | 0.394 | 0.645 |
| | avg | 0.281 | 0.468 | 0.298 | 0.485 | 0.298 | 0.485 | 0.282 | 0.428 | 0.300 | 0.470 | 0.340 | 0.521 | 0.304 | 0.481 | 0.426 | 0.667 | 0.336 | 0.620 | 0.376 | 0.610 | 0.383 | 0.625 |
| Exchange | 96 | 0.200 | 0.083 | 0.211 | 0.092 | 0.235 | 0.090 | 0.213 | 0.092 | 0.208 | 0.088 | 0.207 | 0.087 | 0.205 | 0.088 | 0.367 | 0.256 | 0.234 | 0.107 | 0.278 | 0.148 | 0.218 | 0.088 |
| | 192 | 0.293 | 0.172 | 0.303 | 0.182 | 0.343 | 0.187 | 0.302 | 0.181 | 0.316 | 0.196 | 0.305 | 0.186 | 0.299 | 0.176 | 0.509 | 0.470 | 0.344 | 0.226 | 0.315 | 0.271 | 0.315 | 0.176 |
| | 336 | 0.410 | 0.322 | 0.448 | 0.386 | 0.473 | 0.353 | 0.435 | 0.360 | 0.490 | 0.458 | 0.437 | 0.367 | 0.397 | 0.301 | 0.883 | 1.268 | 0.448 | 0.367 | 0.427 | 0.460 | 0.427 | 0.313 |
| | 720 | 0.693 | 0.855 | 0.767 | 1.004 | 0.761 | 0.934 | 0.928 | 1.442 | 0.841 | 1.318 | 0.706 | 0.874 | 0.714 | 0.901 | 1.068 | 1.767 | 0.746 | 0.964 | 0.695 | 1.195 | 0.695 | 0.839 |
| | avg | 0.399 | 0.358 | 0.432 | 0.416 | 0.453 | 0.391 | 0.470 | 0.519 | 0.464 | 0.515 | 0.414 | 0.379 | 0.404 | 0.367 | 0.707 | 0.940 | 0.443 | 0.416 | 0.429 | 0.519 | 0.414 | 0.354 |

# C  ADDITIONAL HYPER-PARAMETER STUDY

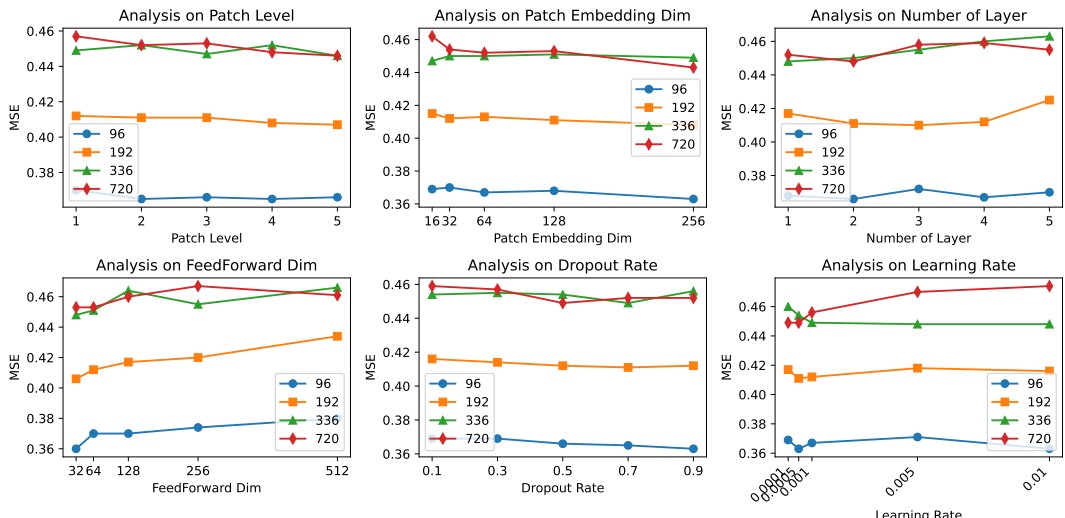

Figure 6: Additional Hyper-parameter Study on ETTm2 with P = {96, 192, 336, 720}

Figure 7: Additional Hyper-parameter Study on ETTh1 with P = {96, 192, 336, 720}

We further conducted hyperparameter experiments on the ETTh1 and ETTm2 dataset, and the results are presented in Figure 6 and Figure 7. Overall, our method demonstrates insensitivity to patch parameters, moderate sensitivity to RHR-MLP parameters, and high sensitivity to the learning rate, consistent with findings from prior parameter analyses. For shorter prediction lengths, the model tends to utilize fewer layers and smaller widths, while for longer prediction lengths, it requires more layers, larger widths, and higher patch levels to enhance information capture. The model achieves superior prediction performance at lower learning rates, albeit at the cost of increased training time.

# D  ADDITIONAL ABLATION STUDY

In Table 5, we provide a comprehensive ablation study. In Cases 1-5, we individually remove each hypercomplex space (RHR-MLP), and the results indicate that eliminating any layer results in performance degradation. Higher-dimensional hypercomplex spaces exhibit a more substantial impact on performance. Moreover, since adjacent dimensions can partially compensate for the missing

Table 5: The full results of the ablation study. The check mark (✓) and the wrong mark (x) indicate with and without components.

| Case | Patch | Hyper-Complex Real | Comp | Quat | Octo | Sede | Adaptive Fusion | ETTh1 with P=720 MSE | MAE | Electricity with P=96 MSE | MAE | ETTh1 with P=96 MSE | MAE |
|---|---|---|---|---|---|---|---|---|---|---|---|---|---|
| 1 | ✓ | x | ✓ | ✓ | ✓ | ✓ | ✓ | 0.451 | 0.451 | 0.160 | 0.246 | 0.367 | 0.384 |
| 2 | ✓ | ✓ | x | ✓ | ✓ | ✓ | ✓ | 0.451 | 0.452 | 0.156 | 0.244 | 0.369 | 0.387 |
| 3 | ✓ | ✓ | ✓ | x | ✓ | ✓ | ✓ | 0.453 | 0.451 | 0.160 | 0.246 | 0.360 | 0.384 |
| 4 | ✓ | ✓ | ✓ | ✓ | x | ✓ | ✓ | 0.453 | 0.455 | 0.161 | 0.247 | 0.369 | 0.387 |
| 5 | ✓ | ✓ | ✓ | ✓ | ✓ | x | ✓ | 0.454 | 0.456 | 0.166 | 0.252 | 0.369 | 0.386 |
| 6 | ✓ | ✓ | ✓ | ✓ | ✓ | ✓ | x | 0.45 | 0.451 | 0.177 | 0.263 | 0.375 | 0.391 |
| 7 | ✓ | ✓ | | | | | - | 0.488 | 0.483 | 0.235 | 0.314 | 0.368 | 0.392 |
| 8 | ✓ | | ✓ | | | | - | 0.469 | 0.469 | 0.214 | 0.292 | 0.369 | 0.391 |
| 9 | ✓ | | | ✓ | | | - | 0.461 | 0.461 | 0.202 | 0.281 | 0.369 | 0.393 |
| 10 | ✓ | | | | ✓ | | - | 0.458 | 0.457 | 0.182 | 0.266 | 0.369 | 0.394 |
| 11 | ✓ | | | | | ✓ | - | 0.457 | 0.454 | 0.180 | 0.263 | 0.369 | 0.394 |
| 12 | x | ✓ | ✓ | ✓ | ✓ | ✓ | ✓ | 0.455 | 0.456 | 0.164 | 0.250 | 0.376 | 0.396 |
| 13 | ✓ | x | ✓ | ✓ | ✓ | ✓ | x | 0.452 | 0.451 | 0.178 | 0.263 | 0.374 | 0.395 |
| 14 | ✓ | ✓ | x | ✓ | ✓ | ✓ | x | 0.451 | 0.452 | 0.178 | 0.262 | 0.372 | 0.393 |
| 15 | ✓ | ✓ | ✓ | x | ✓ | ✓ | x | 0.45 | 0.451 | 0.176 | 0.259 | 0.366 | 0.386 |
| 16 | ✓ | ✓ | ✓ | ✓ | x | ✓ | x | 0.451 | 0.452 | 0.170 | 0.255 | 0.368 | 0.390 |
| 17 | ✓ | ✓ | ✓ | ✓ | ✓ | x | x | 0.45 | 0.453 | 0.175 | 0.259 | 0.365 | 0.391 |
| | ✓ | ✓ | ✓ | ✓ | ✓ | ✓ | ✓ | **0.449** | **0.449** | **0.152** | **0.239** | **0.359** | **0.380** |

spatial information, the performance does not decline significantly when a single hypercomplex layer is removed. In contrast, Cases 7-11 utilize only a single hypercomplex space linear layer, and the notable performance decline emphasizes the importance of integrating multiple hypercomplex linear layers. Each hypercomplex space inherently learns distinct temporal patterns. In Case 12, the removal of Multi-Level Patch Embedding leads to a performance drop, highlighting the necessity of incorporating temporal information of varying lengths, which enables RHR-MLP to model multi-period temporal features. In Case 6, the elimination of the Adaptive Fusion mechanism, where predictions from all spaces are fused with equal weights, results in a significant performance decline, underscoring the critical role of Adaptive Fusion. In Case 13-17, We further compare the ablation of each RHR-MLP in the absence of Adaptive Fusion, and the conclusions remain consistent: adjacent hypercomplex spaces can partially compensate for the missing space, causing a performance drop but not a drastic decline.

Table 6: Ablation and baseline comparison on ETT benchmarks. Lower is better. 'HNTanh' is our full method. Best per *(dataset, P, metric)* is **bold**; ties are bolded for all winners.

| Method | $P$ | ETTh1 MAE | MSE | ETTh2 MAE | MSE | ETTm1 MAE | MSE | ETTm2 MAE | MSE |
|---|---|---|---|---|---|---|---|---|---|
| MLP | 96 | 0.385 | 0.370 | 0.331 | 0.284 | 0.343 | 0.307 | 0.252 | 0.173 |
| *w/o Activation* | 96 | 0.383 | 0.372 | 0.331 | 0.285 | 0.341 | 0.313 | 0.253 | 0.175 |
| *w/o Tanh* | 96 | 0.388 | 0.366 | 0.334 | 0.286 | 0.343 | 0.306 | 0.252 | 0.174 |
| *w/o Dividenorm* | 96 | 0.385 | 0.372 | 0.334 | 0.289 | **0.337** | 0.310 | 0.251 | 0.173 |
| **HNTanh (Ours)** | 96 | **0.380** | **0.359** | **0.326** | **0.279** | **0.337** | **0.305** | **0.249** | **0.170** |
| MLP | 192 | 0.413 | 0.416 | 0.381 | 0.364 | 0.369 | 0.362 | **0.292** | 0.236 |
| *w/o Activation* | 192 | 0.412 | 0.420 | 0.387 | 0.374 | 0.369 | 0.365 | 0.297 | 0.238 |
| *w/o Tanh* | 192 | 0.418 | **0.410** | 0.390 | 0.372 | 0.367 | **0.352** | 0.295 | 0.239 |
| *w/o Dividenorm* | 192 | 0.414 | 0.418 | 0.384 | 0.369 | **0.365** | 0.362 | 0.294 | 0.237 |
| **HNTanh (Ours)** | 192 | **0.409** | 0.407 | **0.379** | **0.359** | 0.367 | 0.356 | **0.292** | **0.231** |

We ablate the proposed hypercomplex activation by disentangling its two components—tanh on the modulus and $p$-norm normalization—and compare against removing activations entirely. On ETT benchmarks, "w/o activation" (only hypercomplex linear layers), "w/o tanh" ($y_i = \frac{x_i}{\|x\|_p}$), and "w/o dividenorm" ($y_i = x_i \tanh(\|x\|_p)$) each show occasional wins on single datasets (e.g., ETTm1), but the full HNTanh consistently yields the best aggregate performance. Theoretically, for the unnormalized variant $y_i = x_i \tanh(\|x\|_p)$, the Jacobian entries shrink as $\|x\|_p \to 0$ (diagonal $\tanh(n) + |x_i|^p (1 - \tanh^2 n) n^{1-p}$, off-diagonal $x_i |x_j|^{p-1} (1 - \tanh^2 n) n^{1-p}$, induction see

Appendix H.2), producing vanishing gradients; adding normalization bounds the input norm and stabilizes gradient flow, matching the empirical gains. We further test a "dimensional expansion" control that pads real inputs (zero-padding/MLP) and trains a standard MLP: despite similar parameter counts, it underperforms, indicating that hypercomplex algebras contribute more than mere width—their structured multiplication/phase coupling provides a superior inductive bias for frequency decomposition and multichannel interactions.

## E  ADDITIONAL EFFICIENCY EXPERIMENTS

We conducted an efficiency evaluation of our model alongside a series of related models. The tests were performed on a single NVIDIA A5000 GPU, with the CPU being an Intel(R) Xeon(R) Gold 6326 @ 2.90GHz. The experimental results on the ETTh1 dataset are presented in Table 7, while those on the solar-energy dataset are shown in Table 8.

Table 7: Comparison of model performance and resource usage in ETTh1 dataset

| Model Name | s/epoch | Parameters | GPU Memory (MB) | MSE | MAE |
|---|---|---|---|---|---|
| Numerion | 0.48 | 904,309 | 700.4 | 0.360 | 0.380 |
| TimeMixer | 0.71 | 75,497 | 1053.76 | 0.377 | 0.388 |
| iTransformer | 0.20 | 224,224 | 156.68 | 0.378 | 0.393 |
| DLinear | 0.10 | 18,624 | 29.18 | 0.379 | 0.386 |
| PaiFilter | 0.10 | 49,614 | 27.17 | 0.380 | 0.390 |
| PatchTST | 0.17 | 21,040 | 173.95 | 0.381 | 0.386 |
| TimeXer | 0.60 | 1,390,944 | 687.13 | 0.384 | 0.394 |
| TimesNet | 9.94 | 605,479 | 10103.31 | 0.399 | 0.404 |
| PatchMLP | 0.26 | 2,470,830 | 506.18 | 0.399 | 0.401 |
| Informer | 0.79 | 422,407 | 1106.81 | 1.204 | 0.800 |

Table 8: Comparison of model performance and resource usage in solar-energy dataset

| Model Name | s/epoch | Parameters | GPU Memory (MB) | MSE | MAE |
|---|---|---|---|---|---|
| Numerion | 53.53 | 1779749 | 2111.33 | 0.217 | 0.250 |
| TimeMixer | 50.26 | 76537 | 2665.77 | 0.236 | 0.254 |
| PaiFilter | 1.36 | 49874 | 179.23 | 0.245 | 0.254 |
| iTransformer | 4.17 | 224224 | 374.72 | 0.246 | 0.255 |
| PatchTST | 6.26 | 21040 | 353.62 | 0.246 | 0.258 |
| TimeXer | 40.66 | 1424224 | 1746.3 | 0.248 | 0.256 |
| TimesNet | 59.76 | 613929 | 1638.73 | 0.277 | 0.296 |
| PatchMLP | 13.3 | 2508530 | 866.04 | 0.282 | 0.280 |
| Informer | 9.68 | 539017 | 303.38 | 0.298 | 0.298 |
| DLinear | 0.95 | 18624 | 151.84 | 0.303 | 0.323 |

The results indicate that, although our model achieves competitive performance, it occupies a middle ground in terms of GPU memory consumption, parameter count, and computational time. Compared to several recent mainstream models, our computational overhead is relatively higher; however, it remains within an acceptable and practical range.

In fact, the core architecture of our model was originally designed with only five linear layers. The increase in parameter count primarily stems from limitations in the underlying PyTorch framework, which lacks native support for high-dimensional hypercomplex operations. As a result, each hypercomplex linear layer is implemented as a pair of real-valued linear layers, effectively simulating hypercomplex computations using real operations. This leads to a substantial increase in the reported parameter count—from five layers to the equivalent of 63 real-valued layers.

The relatively large memory footprint is attributed to the design of multiplication matrices used to simulate hypercomplex multiplications. To accelerate hypercomplex multiplication, we precompute and store multiplication matrices, effectively trading additional memory (up to 16-fold increase in

quadratic space for 16-dimensional hypercomplex numbers) for improved computational speed. This design choice constitutes the primary source of memory overhead.

In terms of computation time, while we reduced the complexity of hypercomplex multiplication from quadratic matrix multiplications to a single matrix multiplication step, the absence of native data type support at the low-level computational layer still results in slower operations. Additionally, the use of a norm tanh activation function, while essential for ensuring stable transformations in the hypercomplex space, inherently incurs higher computational overhead compared to standard activation functions. This inefficiency is further amplified in the hypercomplex setting. Together, these factors contribute to the observed computational overhead.

Nevertheless, despite these inherent challenges posed by the framework and mathematical formulation, the overall efficiency of the model remains tolerable for practical use. Moreover, the architecture leaves room for further optimization, especially with potential future support for hypercomplex data types at the framework level.

## F  ERROR BARS

Table 9: Standard deviation across all datasets

| ETTh1 | | | ETTh2 | | | ETTm1 | | |
|---|---|---|---|---|---|---|---|---|
| | **MSE** | **MAE** | | **MSE** | **MAE** | | **MSE** | **MAE** |
| 96 | 0.380±0.001 | 0.359±0.001 | 96 | 0.326±0.000 | 0.279±0.001 | 96 | 0.337±0.002 | 0.305±0.001 |
| 192 | 0.409±0.001 | 0.407±0.000 | 192 | 0.379±0.001 | 0.359±0.000 | 192 | 0.367±0.003 | 0.356±0.002 |
| 336 | 0.429±0.001 | 0.444±0.001 | 336 | 0.416±0.002 | 0.406±0.002 | 336 | 0.386±0.002 | 0.38±0.002 |
| 720 | 0.449±0.001 | 0.447±0.002 | 720 | 0.433±0.001 | 0.414±0.001 | 720 | 0.423±0.003 | 0.439±0.002 |
| ETTm2 | | | Weather | | | Solar-Energy | | |
| | **MSE** | **MAE** | | **MSE** | **MAE** | | **MSE** | **MAE** |
| 96 | 0.249±0.000 | 0.170±0.000 | 96 | 0.203±0.004 | 0.159±0.002 | 96 | 0.244±0.003 | 0.209±0.004 |
| 192 | 0.292±0.000 | 0.231±0.001 | 192 | 0.25±0.005 | 0.211±0.002 | 192 | 0.261±0.002 | 0.25±0.005 |
| 336 | 0.330±0.001 | 0.289±0.001 | 336 | 0.289±0.005 | 0.270±0.003 | 336 | 0.270±0.003 | 0.273±0.004 |
| 720 | 0.387±0.001 | 0.390±0.001 | 720 | 0.340±0.005 | 0.345±0.003 | 720 | 0.272±0.004 | 0.274±0.007 |
| Electricity | | | Traffic | | | Exchange | | |
| | **MSE** | **MAE** | | **MSE** | **MAE** | | **MSE** | **MAE** |
| 96 | 0.239±0.002 | 0.152±0.000 | 96 | 0.264±0.005 | 0.441±0.002 | 96 | 0.200±0.002 | 0.083±0.001 |
| 192 | 0.251±0.002 | 0.167±0.001 | 192 | 0.276±0.007 | 0.457±0.003 | 192 | 0.293±0.003 | 0.172±0.001 |
| 336 | 0.270±0.003 | 0.182±0.002 | 336 | 0.282±0.006 | 0.465±0.003 | 336 | 0.270±0.003 | 0.322±0.001 |
| 720 | 0.308±0.003 | 0.224±0.004 | 720 | 0.301±0.007 | 0.507±0.004 | 720 | 0.693±0.004 | 0.855±0.002 |

In Table 9, we present the standard deviations calculated from three repeated experiments on Numerion, highlighting its exceptional stability as a model. The results consistently exhibit low standard deviations across all datasets, with particularly notable performance on the ETT dataset. This underscores Numerion's robustness and reliability in time series forecasting tasks.

## G  HIGH EFFICIENT HYPERCOMPLEX LINEAR LAYERS

In this paper, The defination of **hypercomplex algebra** based on the **Cayley-Dickson algebra system**. The Cayley-Dickson construction provides a general recursive method for generating higher-dimensional algebras by repeatedly doubling lower-dimensional ones.

We begin with the complex numbers, which can be viewed as a two-dimensional algebra $\mathbb{R} \times \mathbb{R}$ over the real field $\mathbb{R}$, where each element is represented as a pair of real numbers $(\alpha_1, \beta_1)$, $(\alpha_2, \beta_2)$. The addition of these elements follows the natural coordinate-wise rule:

$$(\alpha_1, \beta_1) + (\alpha_2, \beta_2) = (\alpha_1 + \alpha_2, \beta_1 + \beta_2)$$

Under the guidance of the Cayley-Dickson construction, the multiplication is defined by:

$$(\alpha_1, \beta_1) \times (\alpha_2, \beta_2) = (\alpha_1 \alpha_2 - \beta_2^* \beta_1, \; \beta_2 \alpha_1 + \beta_1 \alpha_2^*)$$

where $d^*$ denotes the conjugate of $d$. Since $d$ is real in this base case, the conjugate has no effect. It is straightforward to verify that this multiplication satisfies the bilinearity property, thus constituting a valid algebraic structure. By iterating this construction, we can systematically define multiplication rules for higher-dimensional algebras such as quaternions, octonions, and beyond.

Theoretically, the time complexity of our layer is $O(T \cdot E)$, which is linear with respect to the time series length $T$. However, the recursive nature of Cayley-Dickson multiplication means the computational cost scales quadratically with the algebra dimension $n$. In actual running, specifically for high-order algebras like Sedenions ($n = 16$), this results in a very large constant multiplier (approx. $63\times$ FLOPs vs. scalar), making naive sequential implementation extremely slow. Therefore, to optimize this "large constant" and ensure practical efficiency, we propose the following vectorized implementation:

In practice, real-number computations enjoy significant efficiency advantages in frameworks like PyTorch. Therefore, in our implementation, we represent hypercomplex numbers as real-valued vectors, leveraging extra dimensions to encode hypercomplex coefficients. Specifically, we employ the Cayley-Dickson construction to generate two matrices: a **coefficient selection matrix** and a **sign matrix**. The former specifies which coefficients of the operands should be multiplied at each step, while the latter encodes the corresponding signs to apply to the resulting products. During each multiplication operation, these matrices are instantiated using PyTorch tensor operations, enabling efficient vectorized computation.

As an example, suppose $p, q$ is the quaternion numbers, we have quaternion multiplication :

$$
\begin{aligned}
p \times q = &(p_0 q_0 - p_1 q_1 - p_2 q_2 - p_3 q_3) \\
&+ (p_0 q_1 + p_1 q_0 + p_2 q_3 - p_3 q_2)\mathbf{i}_1 \\
&+ (p_0 q_2 - p_1 q_3 + p_2 q_0 + p_3 q_1)\mathbf{i}_2 \\
&+ (p_0 q_3 + p_1 q_2 - p_2 q_1 + p_3 q_0)\mathbf{i}_3
\end{aligned}
\tag{11}
$$

We can explicitly extract both the coefficient selection matrix and the sign matrix corresponding to the Cayley-Dickson construction at each algebraic level. The coefficient selection matrix determines (left below) which elements from the input vectors should participate in each multiplication term, while the sign matrix (right below) encodes the appropriate sign (positive or negative) for each product.

$$
\begin{bmatrix}
0 & 1 & 2 & 3 \\
1 & 0 & 3 & 2 \\
2 & 3 & 0 & 1 \\
3 & 2 & 1 & 0
\end{bmatrix}
\quad
\begin{bmatrix}
1 & -1 & -1 & -1 \\
1 & 1 & 1 & -1 \\
1 & -1 & 1 & 1 \\
1 & 1 & -1 & 1
\end{bmatrix}
$$

By precomputing these matrices, we effectively transform the recursive multiplication process into a fixed matrix operation. Specifically, quaternion (or more generally, hypercomplex) multiplication can be reformulated as a linear transformation:

$$\mathbf{y} = S \cdot ((M \odot \mathbf{w})\, \mathbf{x})$$

Here, $M$ is the coefficient selection matrix, $S$ is the sign matrix, $\odot$ denotes element-wise multiplication (interpreted as a gather-and-weight process), $\mathbf{x}$ is the input vector, and $\mathbf{w}$ represents the learnable weights or operands. The weight tensor $\mathbf{w}$ is shaped as output_dim $\times$ input_dim $\times$ $n$, where $n$ is the number of hypercomplex coefficients. Let $\mathbf{w}_i^{(n)}$ denote the $i$-th coefficient of the hypercomplex

vector $\mathbf{w}$. The coefficient selection matrix $M \in \mathbb{N}^{n \times n}$ is used to construct the multiplication matrix $A$ shaped as output_dim $\times$ input_dim $\times n \times n$ according to:

$$A_{i,j} = \mathbf{w}_{M[i,j]}^{(n)}$$

In this expression, $\mathbf{w}_{M[i,j]}^{(n)}$ is not a shallow copy but a newly constructed tensor derived from the original $\mathbf{w}^{(n)}$ via an index-based gather operation. Despite creating a new tensor, the autograd mechanism retains the connection to $\mathbf{w}^{(n)}$ in the computational graph. As a result, multiplying the input $\mathbf{x}$ with matrix $A$ remains algebraically equivalent to hypercomplex multiplication with $\mathbf{w}$, and gradients are correctly and efficiently propagated back to the original $j$-th coefficient of $\mathbf{w}^{(n)}$ during backpropagation, without introducing redundancy.

This approach enables us to express hypercomplex multiplication as a unified matrix multiplication step. Although it necessitates temporarily building the multiplication matrix for each computation—introducing some additional CPU/GPU overhead—it substantially accelerates the overall computation by reducing the number of separate operations and fully utilizing hardware-optimized matrix kernels. Consequently, this parallelized strategy effectively neutralizes the "large constant" cost identified in the complexity analysis above, allowing the model to maintain linear scaling without the wall-clock latency penalty.

The multiplication rules for octonions is provided below:

$$
\begin{bmatrix}
0 & 1 & 2 & 3 & 4 & 5 & 6 & 7 \\
1 & 0 & 3 & 2 & 5 & 4 & 7 & 6 \\
2 & 3 & 0 & 1 & 6 & 7 & 4 & 5 \\
3 & 2 & 1 & 0 & 7 & 6 & 5 & 4 \\
4 & 5 & 6 & 7 & 0 & 1 & 2 & 3 \\
5 & 4 & 7 & 6 & 1 & 0 & 3 & 2 \\
6 & 7 & 4 & 5 & 2 & 3 & 0 & 1 \\
7 & 6 & 5 & 4 & 3 & 2 & 1 & 0
\end{bmatrix}
\begin{bmatrix}
1 & -1 & -1 & -1 & -1 & -1 & -1 & -1 \\
1 & 1 & -1 & 1 & -1 & 1 & 1 & -1 \\
1 & 1 & 1 & -1 & -1 & -1 & 1 & 1 \\
1 & -1 & 1 & 1 & -1 & 1 & -1 & 1 \\
1 & 1 & 1 & 1 & 1 & -1 & -1 & -1 \\
1 & -1 & 1 & -1 & 1 & 1 & 1 & -1 \\
1 & -1 & -1 & 1 & 1 & -1 & 1 & 1 \\
1 & 1 & -1 & -1 & 1 & 1 & -1 & 1
\end{bmatrix}
$$

# H    THEORETICAL SUPPLEMENT

## H.1    GRADIENT OF ACTIVATION FUNCTION

Although the HNTanh activation function in this paper operates on hypercomplex numbers, it can equivalently be viewed as an activation function for multivariate real vectors. Given an input vector $x \in \mathbb{R}^m$, the output after applying the activation function is $y = f(x) \in \mathbb{R}^m$.

Our goal is to derive the Jacobian matrix $J = \frac{\partial y}{\partial x}$. Suppose we use $x_i$ to represent the $i$-th coefficient of the vector $x$. Due to the inherent symmetry of the function, it suffices to compute the diagonal entries $\frac{\partial y_i}{\partial x_i}$ and the off-diagonal entries $\frac{\partial y_i}{\partial x_j}$ for $i \neq j$.

The activation function is defined coefficient-wise as:

$$y_i = \frac{x_i}{\|x\|_p} \tanh(\|x\|_p),$$

where $\|x\|_p$ denotes the $p$-norm:

$$\|x\|_p = \left( \sum_{k=1}^{m} |x_k|^p \right)^{1/p}.$$

Let $n := \|x\|_p$ for brevity.

We differentiate $y_i$ with respect to $x_i$:

$$\frac{\partial y_i}{\partial x_i} = \frac{\partial}{\partial x_i} \left( \frac{x_i}{n} \tanh(n) \right).$$

Applying the product derivative rule:

$$\frac{\partial y_i}{\partial x_i} = \frac{\tanh(n)}{n} + x_i \frac{\partial}{\partial x_i}\left(\frac{\tanh(n)}{n}\right).$$

Next, by the chain rule:

$$\frac{\partial}{\partial x_i}\left(\frac{\tanh(n)}{n}\right) = \frac{\partial n}{\partial x_i}\frac{\partial}{\partial n}\left(\frac{\tanh(n)}{n}\right).$$

Compute the derivative with respect to $n$:

$$\frac{\partial}{\partial n}\left(\frac{\tanh(n)}{n}\right) = \frac{n(1 - \tanh^2(n)) - \tanh(n)}{n^2}.$$

Compute the derivative of $n$ with respect to $x_i$:

$$\frac{\partial n}{\partial x_i} = |x_i|^{p-1}\operatorname{sign}(x_i)n^{1-p}.$$

Therefore, the final expression for $\frac{\partial y_i}{\partial x_i}$ is:

$$\boxed{\frac{\partial y_i}{\partial x_i} = \frac{\tanh(n)}{n} + |x_i|^p\left(\frac{n(1 - \tanh^2(n)) - \tanh(n)}{n^2}\right)n^{1-p}.}$$

Similarly, differentiating $y_i$ with respect to $x_j$:

$$\frac{\partial y_i}{\partial x_j} = x_i \frac{\partial}{\partial x_j}\left(\frac{\tanh(n)}{n}\right).$$

Again applying the chain rule:

$$\frac{\partial y_i}{\partial x_j} = x_i \frac{\partial n}{\partial x_j}\frac{\partial}{\partial n}\left(\frac{\tanh(n)}{n}\right).$$

We already computed $\frac{\partial}{\partial n}\left(\frac{\tanh(n)}{n}\right)$. For $\frac{\partial n}{\partial x_j}$:

$$\frac{\partial n}{\partial x_j} = |x_j|^{p-1}\operatorname{sign}(x_j)n^{1-p}.$$

Thus:

$$\boxed{\frac{\partial y_i}{\partial x_j} = x_i|x_j|^{p-1}\operatorname{sign}(x_j)\left(\frac{n(1 - \tanh^2(n)) - \tanh(n)}{n^2}\right)n^{1-p}.}$$

The Jacobian matrix entries are thus fully determined, Diagonal entries by the formula above for $\frac{\partial y_i}{\partial x_i}$, Off-diagonal entries by the formula above for $\frac{\partial y_i}{\partial x_j}$ with $i \neq j$.

This derivation completes the gradient calculation of the HNTanh activation function.

## H.2 GRADIENT OF HYPERCOMPLEX NUERAL NETWORK

In the context of training hypercomplex neural networks, it is essential to demonstrate that the results obtained through real-valued gradient descent are valid within the hypercomplex domain. Our justification is twofold: first, we establish the mechanical validity of the gradient via the Jacobian matrix; second, we provide the theoretical grounding for optimization using **Liouville's Theorem** and **GHR Calculus**.

**Mechanical Validity: The Real-Valued Jacobian**    First, the network is demonstrably trainable when treated as a real-differentiable function. By viewing hypercomplex multiplication as a linear transformation on real-valued vectors, the entire network inherits differentiability in the real sense.

To verify this, we compare the gradient computed in our network against the true Jacobian matrix derived from real differentiation. Taking a quaternion-based linear layer as an example:

$$F : \mathbb{R}^4 \to \mathbb{R}^4, \quad F(c^{(4)}) = W \times c^{(4)} + B \tag{12}$$

Here, the input $c^{(4)}$, weights $W$, and bias $B$ are quaternions treated as vectors of four real coefficients. Multiplication follows standard quaternion rules (Equation 11). Computing the Jacobian of the output $F(c^{(4)})$ with respect to $c^{(4)}$ yields:

$$J_W = \begin{bmatrix} W_0 & -W_1 & -W_2 & -W_3 \\ W_1 & W_0 & W_3 & -W_2 \\ W_2 & -W_3 & W_0 & W_1 \\ W_3 & W_2 & -W_1 & W_0 \end{bmatrix} \tag{13}$$

This structure reveals that each coefficient of $W$ contributes to the output via a linear transformation consistent with the coefficient matrices defined in our implementation (Appendix G). Furthermore, it is not difficult to extend this analysis to higher-dimensional hypercomplex numbers. Consequently, the gradients derived from standard backpropagation are exactly equivalent to this Jacobian, confirming that real-valued optimization methods are mechanically applicable.

**Theoretical Validity: Liouville's Theorem and GHR Calculus**    While the network is mechanically trainable, a theoretical distinction must be made regarding hypercomplex differentiability. Our implementation does not strictly satisfy the generalized **Cauchy-Riemann–Fueter (CRF) equations**. Still taking quaternion function $f$ as an example, strict differentiability requires:

$$Df := \left( \frac{\partial}{\partial X_0} + \mathbf{i}_1 \frac{\partial}{\partial X_1} + \mathbf{i}_2 \frac{\partial}{\partial X_2} + \mathbf{i}_3 \frac{\partial}{\partial X_3} \right) f = 0 \tag{14}$$

However, enforcing this condition is detrimental to neural network design. According to **Liouville's Theorem** for hypercomplex analysis (Sudbery, 1979), any bounded function that strictly satisfies the CRF conditions is necessarily constant. Even in relaxed definitions, the class of functions that are strictly hypercomplex-differentiable is restricted to linear or affine transformations. This restriction would render the construction of deep, non-linear neural networks impossible.

Therefore, we do not seek strict hypercomplex differentiability. Instead, our optimization framework relies on **GHR (Generalized Hamilton-Real) Calculus**.(Xu et al., 2015) GHR calculus generalizes the complex Wirtinger calculus to quaternions and other algebras, providing a formal framework for minimizing real-valued loss functions (which are non-holomorphic by definition) with respect to hypercomplex parameters.

By employing GHR calculus, we treat the hypercomplex weights and their conjugates as independent variables during differentiation. This theoretically validates our approach: the "real-valued gradient descent" we employ is, in rigorous terms, an implementation of GHR calculus, allowing for effective training without being constrained by the rigidity of Liouville's Theorem.

## I    ACTIVATION MECHANISMS IN HYPERCOMPLEX NEURAL NETWORKS

Although hypercomplex networks remain a niche area, researchers have increasingly explored their potential in various deep learning tasks. One major limitation hindering their broader success is the lack of effective activation functions tailored to hypercomplex spaces. Prior approaches often apply real-valued activation functions independently to each component, which breaks the inherent symmetry of hypercomplex representations and restricts expressive capacity during training. Notably, the modulus—shared across all hypercomplex systems—possesses desirable mathematical properties.

This motivates us to develop activation functions that operate on the modulus directly, aiming to preserve structural integrity and improve learning efficiency.

In this paper, we propose HNTanh, an activation function that leverages the modulus property of hypercomplex numbers and treats the hypercomplex input holistically rather than coefficient-wise. Specifically, given a hypercomplex input x, the activation function is defined as:

$$\text{HNTanh}(c^{(n)}) = \frac{c^{(n)}}{\|c^{(n)}\|_p} \cdot \tanh(\|c^{(n)}\|_p)$$

where

$$\tanh(x) = \frac{e^x - e^{-x}}{e^x + e^{-x}},$$

$$\|c^{(n)}\|_p = \left( \sum_i |c_i^{(n)}|^p \right)^{1/p},$$

$\|c^{(n)}\|_p$ denotes the p-norm of n-dimensional hypercomplex number $c^{(n)}$. In this work, we set $p = 6$. We visualize the state space of the activation function on a two-dimensional plane. Specifically, Figure 8 shows the activation function itself; Figure 9 depicts the gradient of the activation function with respect to the modulus; Figures 10 and 11 illustrate the gradients with respect to the first and second coefficients, respectively. These visualizations demonstrate that our activation function exhibits stronger overall symmetry and maintains a smoother gradient distribution across most regions of the space.

## I.1 WHY DO WE CHOOSE SIX-NORM?

To determine the optimal choice of norm, we conducted both visualization and empirical evaluations. Visualization on the complex plane revealed that using a six-norm, as shown in Figure 0, produces pronounced fluctuations and sharper transitions around extreme values. In contrast, lower-order norms, such as the two-norm depicted in Figure 12, exhibit smoother behavior in non-extreme regions. Higher-order norms, including the infinity norm,depicted in Figure 13, introduced excessive extreme points, leading to poorer overall continuity in the function's profile.And from the formula in H.1, we can see that the Jacobian matrix is almost 0 except for the diagonal and the rows and columns where the maximum value elements are located, and the optimization is unstable.

For the experimental evaluation, we trained models on multiple variants of the ETT dataset (ETTh1, ETTh2, and ETTm1) using p-norm values ranging from 1 to 8 and inf, with all other settings kept identical. The results, summarized in Table 10, reveal that due to differences in the data distributions across datasets, it is not feasible to depict a single unified performance curve showing the relationship between the norm value and overall model performance. Nevertheless, it is evident that activation functions with medium to high norm values consistently outperform those with lower norms. Notably, the six-norm, as part of this medium-to-high norm range, achieves relatively strong performance across datasets. Based on these empirical findings and the smoother, more balanced properties observed in the visualization, we adopt the six-norm as our activation function.

## I.2 WHY DO WE CHOOSE TANH?

The norm-based activation function for hypercomplex numbers can effectively induce a symmetric and relatively smooth functional space. However, determining the optimal choice of activation function remains an open question worthy of deeper investigation. Indeed, selecting an appropriate activation function is a highly complex problem. The study of activation functions itself is challenging, as it lacks universally rigorous theoretical frameworks and definitive solutions that generalize well across applications. On the one hand, the performance of an activation function is closely tied to the specific characteristics of the downstream tasks; on the other hand, research on activation functions in hypercomplex spaces remains scarce.

In this work, we therefore build upon well-established activation functions with widespread empirical success. We conducted an analysis of the currently mainstream ReLU-like activation functions and the sigmoid function, and visualized their gradient behavior with respect to the norm (modulus) of

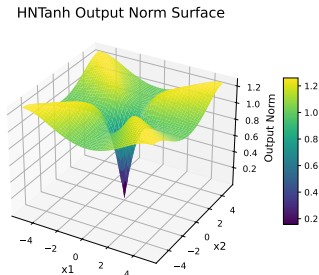

Figure 8: Visualization of the HNTanh activation function over complex numbers.

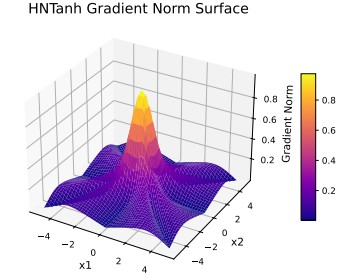

Figure 9: Visualization of the gradient of the norm induced by the HNTanh activation function.

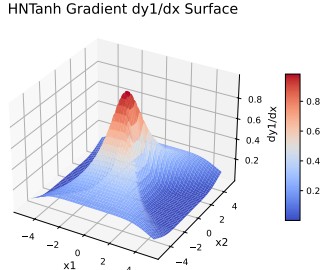

Figure 10: Visualization of the gradient of the real coefficient (Re) in the HNTanh activation function.

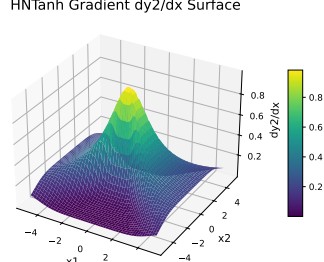

Figure 11: Visualization of the gradient of the imaginary coefficient (Im) in the HNTanh activation function.

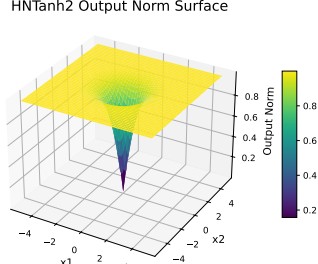

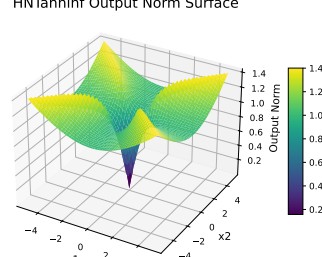

Figure 12: Visualization of the 2-norm tanh activation function space. The function exhibits noticeably over-smoothed field characteristics.

Figure 13: Visualization of the $\infty$-norm tanh activation function space. The function displays clearly sawtooth-like field patterns.

Table 10: Performance of different activation function norms on the ETT dataset, evaluated under identical experimental conditions.

| Order of Norm | ETTh1 | | ETTh2 | | ETTm1 | |
|---|---|---|---|---|---|---|
| | MAE | MSE | MAE | MSE | MAE | MSE |
| 1 | 0.3838 | 0.3629 | 0.3319 | 0.2877 | 0.3415 | 0.3046 |
| 2 | 0.3805 | 0.3604 | 0.3329 | 0.2911 | 0.3425 | 0.3049 |
| 3 | 0.3800 | 0.3604 | 0.3313 | 0.2870 | 0.3424 | 0.3063 |
| 4 | 0.3801 | 0.3609 | 0.3313 | 0.2869 | 0.3419 | 0.3072 |
| 5 | 0.3795 | 0.3599 | 0.3309 | 0.2870 | 0.3412 | 0.3056 |
| 6 | 0.3798 | 0.3602 | 0.3306 | 0.2854 | 0.3401 | 0.3057 |
| 7 | 0.3798 | 0.3600 | 0.3301 | 0.2853 | 0.3405 | 0.3057 |
| 8 | 0.3797 | 0.3597 | 0.3310 | 0.2867 | 0.3398 | 0.3049 |
| inf | 0.3795 | 0.3593 | 0.3306 | 0.2846 | 0.3415 | 0.3069 |

hypercomplex numbers. As illustrated in Figures 14 and 15, unlike the real-valued case where inputs span both positive and negative domains, the modulus is inherently non-negative. Consequently, activation functions in the ReLU family—exemplified here by the SiLU function—exhibit a nearly uniform gradient across the complex plane, effectively activating only the positive half of the input domain. Sigmoid-based activations also suffer from similar limitations under this setting.

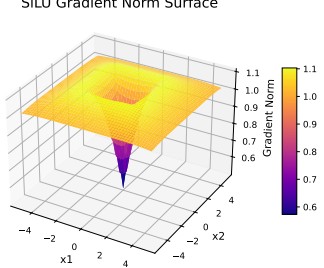

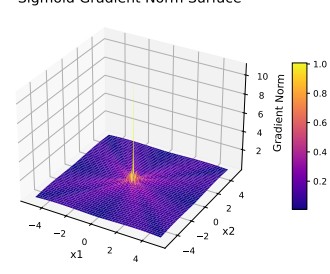

Figure 14: Visualization of activation function gradient under SiLU replacement

Figure 15: Visualization of activation function gradient under Sigmoid replacement

In our validation experiments, we selected several widely used and influential activation functions. Within the Norm-Activation framework, we fixed the overall structure while varying only the activation function, ensuring that all other experimental parameters remained identical. The results are presented in the table below.

Table 11: Performance of different activation function norms on the ETT dataset, evaluated under identical experimental conditions.

| Type of Function | ETTh1 | | ETTh2 | | ETTm1 | |
|---|---|---|---|---|---|---|
| | MAE | MSE | MAE | MSE | MAE | MSE |
| Tanh | 0.3800 | 0.3602 | 0.3337 | 0.2901 | 0.3416 | 0.3068 |
| Sigmoid | 0.3855 | 0.3671 | 0.3366 | 0.2944 | 0.3409 | 0.3041 |
| Relu | 0.3815 | 0.3633 | 0.3316 | 0.2861 | 0.3483 | 0.3234 |
| Mish | 0.3824 | 0.3667 | 0.3324 | 0.2899 | 0.3507 | 0.3337 |
| SiLU | 0.3842 | 0.3695 | 0.3324 | 0.2904 | 0.3438 | 0.3193 |
| LogSigmoid | 0.3993 | 0.3945 | 0.3398 | 0.3068 | 0.3562 | 0.3314 |
| RRelu | 0.3814 | 0.3632 | 0.3319 | 0.2868 | 0.3515 | 0.3362 |

Notably, Tanh achieved the best performance on the ETTh1 dataset and performed comparably well on both ETTh2 and ETTm1. Although it did not attain the top score across every dataset, Tanh consistently delivered the most competitive overall results among all tested activation functions. These findings provide empirical support for the effectiveness of selecting Tanh as the activation function, validating our choice from a practical experimental standpoint.

## J  VISUALIZATION

### J.1  VISUALIZATION OF HYPERCOMPLEX LINEAR PREDICTION RESULTS

We visualized the outputs of each linear block in the trained model under its best-performing parameters. For this analysis, we selected the prediction results of the first channel from the first sample in the ETTh1 test set. Figure18 presents the predictions from five different linear layers in the time domain, while Figure17 shows their corresponding representations in the frequency domain. Figure16 illustrates the weighted average weights assigned to each linear block, which are used to average the outputs of the RHR-MLP.

The results reveal that, under optimal settings, the model implicitly assigns the high-dimensional hypercomplex layer to predict the low-dimensional, low-frequency components of the time series, while the real and complex linear layers focus on capturing the high-dimensional, high-frequency components. This separation enables the model to effectively fit localized, noisy seasonal signals while maintaining accurate predictions of the overall trend. Additionally, the average weight distribution indicates that the real and complex linear layers collectively contribute approximately 40% of the prediction sequence, primarily capturing medium and high-frequency seasonal signals. In contrast, the hypercomplex layers account for about 60% of the results, with the hexadecimal linear layer alone supporting nearly 40% of the predictions, while the remaining hypercomplex layers play an auxiliary role in refining the overall trend.

Interestingly, while this decomposition behavior resembles the effects of classical time series decomposition, it emerges naturally through model training. Unlike artificially designed decomposition models, it is not constrained by the limitations of fixed-form decomposition. This statistical, machine learning–driven decomposition ability enables the model to adapt flexibly to different datasets, ultimately contributing to its strong predictive performance across multiple benchmarks. It also provides a glimpse into the nature of the model's low- and high-frequency features in learning time series data. We will explore this further in the next paragraph, and the section of quantitative analysis of these features.

## J.2 VISUALIZATION RESULTS WITH OTHER TRAINING SETTING

In order to better explore the inspiration brought by this method, We conducted visualization experiments under alternative experimental settings to further validate our findings. Figures 20 and 21 illustrate the visualization effects of the ETTh1 dataset under non-optimal parameters, while Figure 19 presents the corresponding average weight distribution. The results indicate that even under non-optimal configurations, the model consistently employs high-dimensional hypercomplex layers as the primary mechanism for capturing low-frequency information, while relying on real and complex layers to capture high-frequency signals. However, under these suboptimal conditions, the low-dimensional layers exhibit a relatively poor learning effect on the high-frequency components, with their corresponding weight proportion significantly lower than the 40% observed under optimal configurations. Consequently, their contribution to the overall decomposition is minimal, leading to subpar model prediction results.

We also performed visualization experiments on other datasets.Figures 23, 24, and 22 present the visualization of specific channels on the ETTh2 dataset, while Figures 29, 30, 28, 26, 27, and 25 show the corresponding visualizations for the ETTm2, ETTm1 dataset. Although these datasets exhibit more complex high-frequency signals, the dominance of hypercomplex numbers in the learned weights, along with the observation that different hypercomplex layers capture distinct low-frequency features, suggests that the model primarily focuses on learning low-frequency trend characteristics. Based on these experiments, we propose the following hypothesis: the model's inability to accurately predict low-frequency signals limits its capacity to further analyze high-frequency components, thereby constraining overall performance improvement. While this conclusion may not be entirely definitive—given the inherent limitations of the datasets and model architecture—it nonetheless offers valuable insights for further enhancing our time-series prediction model.

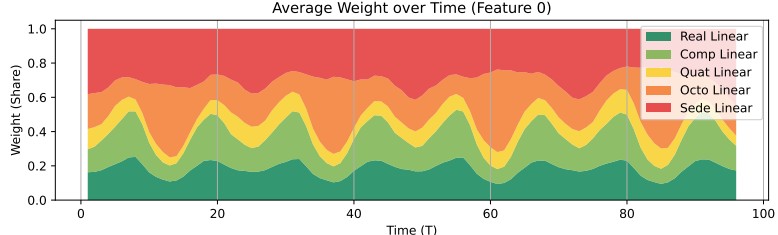

Figure 16: Visualization of weights of MLP on ETTh1

Figure 17: Visualization of MLP prediction in frequency domain on ETTh1

Figure 18: Visualization of MLP prediction in time domain on ETTh1

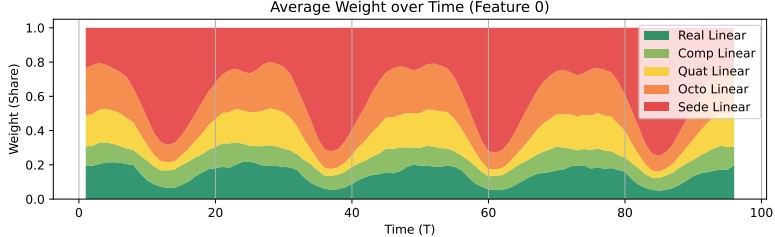

Figure 19: Visualization of weights of MLP on ETTh1 on non-optimal parameters

Figure 20: Visualization of MLP prediction in frequency domain on ETTh1(non-optimal)

Figure 21: Visualization of MLP prediction in time domain on ETTh1(non-optimal)

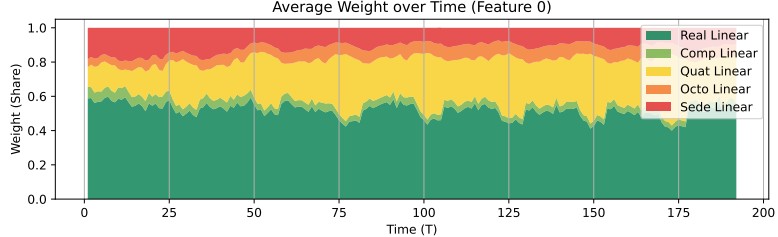

Figure 22: Visualization of weights of MLP on ETTh2

Figure 23: Visualization of MLP prediction in frequency domain on ETTh2

Figure 24: Visualization of MLP prediction in time domain on ETTh2

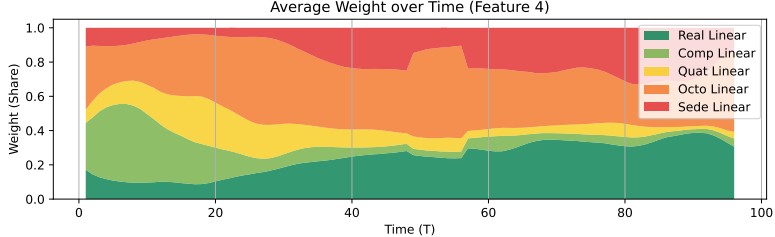

Figure 25: Visualization of weights of MLP on ETTm1

Figure 26: Visualization of MLP prediction in frequency domain on ETTm1

Figure 27: Visualization of MLP prediction in time domain on ETTm1

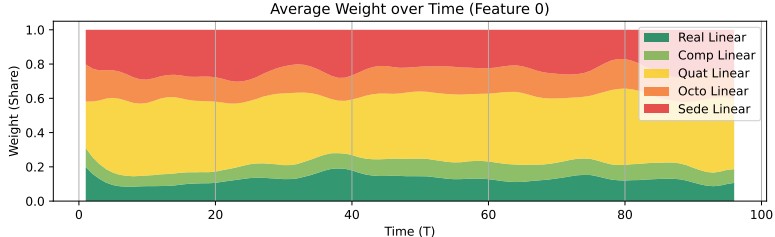

Figure 28: Visualization of weights of MLP on ETTm2

Figure 29: Visualization of MLP prediction inFigure 30: Visualization of MLP prediction in frequency domain on ETTm2                        time domain on ETTm2

## K  QUANTITATIVE STATISTICAL ANALYSIS TO THE DECOMPOSITION METHODS

To further validate the characteristics of the decomposition approach described in the previous section and to derive concrete insights from the unique decomposition learned by the model, we conducted extensive quantitative analyses on the ETT dataset. Specifically, we computed three key metrics for the predictions made by real, complex, and hypercomplex linear layers in the frequency domain: the **Proportion of Variance Explained (PVE)**Fraser (1965), the **Mean Absolute Frequency (MAF)**Haugen et al. (2015), and the **Pearson correlation coefficient**Rahadian et al. (2023) between predictions and ground truth in the time domain.

PVE, a classical metric in variance analysis, measures the proportion of variance in the dependent variable that is explained by the independent variable. Letting $Y_f$ denote the ground truth and $\hat{Y}_f$ denote the predicted value at frequency $f$, the PVE is mathematically defined as:

$$\text{PVE}_f = 1 - \frac{\text{Var}(Y_f - \hat{Y}_f)}{\text{Var}(Y_f)}$$

A PVE value closer to 1 indicates a higher explanatory power. In this study, we partitioned the frequency domain into intervals to analyze the PVE across different frequency bands, thereby revealing how well each RHR-MLP fits different frequency regions.

The MAF metric reflects the frequency specificity of the predictions by the linear layers. A lower MAF indicates that the predictions are dominated by lower-frequency components, whereas a higher MAF suggests a greater presence of high-frequency components in the predictions. The MAF in this study is computed as:

$$\text{MAF} = \frac{\sum_f |f| \cdot P(f)}{\sum_f P(f)}$$

where $P(f)$ denotes the power at frequency $f$.

Table 12: Mean PVE per RHR-MLP per frequency bin on channel 0 of ETTh1, with corresponding Mean Absolute Frequency (MAF) and Pearson correlation coefficient.

|      | [0.00-0.01] | [0.01-0.04] | [0.04-0.09] | [0.09-0.14] | [0.14-0.20] | [0.20-0.27] | [0.27-0.50] | **MAF** | **Pearson** |
|------|------|------|------|------|------|------|------|------|------|
| Real | 0.0004 | 0.0156 | 0.0377 | 0.0091 | 0.3040 | 0.1878 | 0.4311 | 0.241223 | 0.097605 |
| Comp | 0.0432 | 0.0075 | 0.9196 | 0.0103 | 0.0065 | 0.0083 | 0.0045 | 0.082320 | 0.382733 |
| Quat | 0.2199 | 0.0604 | 0.6982 | 0.0138 | 0.0035 | 0.0017 | 0.0023 | 0.037663 | 0.521406 |
| Octo | 0.1036 | 0.0230 | 0.7892 | 0.0740 | 0.0037 | 0.0041 | 0.0024 | 0.051302 | 0.631457 |
| Sede | 0.1000 | 0.0116 | 0.8445 | 0.0362 | 0.0026 | 0.0028 | 0.0022 | 0.046143 | 0.688479 |

Table 13: Mean PVE per RHR-MLP per frequency bin on channel 1 of ETTh1, with corresponding Mean Absolute Frequency (MAF) and Pearson correlation coefficient.

|      | [0.00-0.01] | [0.01-0.04] | [0.04-0.09] | [0.09-0.14] | [0.14-0.20] | [0.20-0.27] | [0.27-0.50] | **MAF** | **Pearson** |
|------|------|------|------|------|------|------|------|------|------|
| Real | 0.0004 | 0.0087 | 0.0208 | 0.0066 | 0.277 | 0.1997 | 0.4685 | 0.257347 | 0.114217 |
| Comp | 0.0612 | 0.0107 | 0.8371 | 0.0173 | 0.0153 | 0.0428 | 0.0155 | 0.089623 | 0.304849 |
| Quat | 0.326 | 0.0784 | 0.5738 | 0.0146 | 0.0034 | 0.0016 | 0.0022 | 0.032386 | 0.45547 |
| Octo | 0.1592 | 0.0192 | 0.7165 | 0.0797 | 0.007 | 0.0121 | 0.0062 | 0.05182 | 0.479334 |
| Sede | 0.168 | 0.0152 | 0.7587 | 0.0385 | 0.0057 | 0.0078 | 0.006 | 0.045574 | 0.565875 |

The third metric, the Pearson correlation coefficient, measures the overall similarity between the predicted and ground truth signals in the time domain. On typical time series datasets, a Pearson coefficient closer to 1 indicates that the predicted sequence better aligns with the overall trend of the true sequence.

Tables 12,13 present the results of our quantitative analyses on channels 0 and 1 of the ETTh1 dataset. The corresponding MAF and Pearson correlation coefficients are reported alongside the interval-based PVE values. From these results, we observe similar patterns to those discussed in

Table 14: Mean PVE per RHR-MLP per frequency bin on channel 0 of ETTm1, with corresponding Mean Absolute Frequency (MAF) and Pearson correlation coefficient.

| | [0.00-0.01] | [0.01-0.04] | [0.04-0.09] | [0.09-0.14] | [0.14-0.20] | [0.20-0.27] | [0.27-0.50] | MAF | Pearson |
|---|---|---|---|---|---|---|---|---|---|
| Real | 0.2485 | 0.7226 | 0.0237 | 0.0033 | 0.0007 | 0.0004 | 0.0008 | 0.011422 | 0.446613 |
| Comp | 0.4764 | 0.4804 | 0.0315 | 0.0073 | 0.0019 | 0.0007 | 0.0017 | 0.010222 | 0.037417 |
| Quat | 0.3179 | 0.4730 | 0.1607 | 0.0172 | 0.0135 | 0.0044 | 0.0133 | 0.026225 | 0.408534 |
| Octo | 0.0173 | 0.9675 | 0.0139 | 0.0006 | 0.0002 | 0.0001 | 0.0003 | 0.013652 | 0.788544 |
| Sede | 0.1989 | 0.7693 | 0.0268 | 0.0023 | 0.0010 | 0.0005 | 0.0012 | 0.012790 | 0.678040 |

Table 15: Mean PVE per RHR-MLP per frequency bin on channel 3 of ETTm1, with corresponding Mean Absolute Frequency (MAF) and Pearson correlation coefficient.

| | [0.00-0.01] | [0.01-0.04] | [0.04-0.09] | [0.09-0.14] | [0.14-0.20] | [0.20-0.27] | [0.27-0.50] | MAF | Pearson |
|---|---|---|---|---|---|---|---|---|---|
| Real | 0.1378 | 0.8323 | 0.0222 | 0.005 | 0.001 | 0.0006 | 0.0011 | 0.013584 | 0.208974 |
| Comp | 0.6085 | 0.3462 | 0.0315 | 0.0089 | 0.0022 | 0.0007 | 0.0019 | 0.009141 | 0.028703 |
| Quat | 0.4405 | 0.3668 | 0.1465 | 0.0181 | 0.0117 | 0.0043 | 0.0122 | 0.023216 | 0.305707 |
| Octo | 0.0069 | 0.9701 | 0.0205 | 0.0011 | 0.0004 | 0.0002 | 0.0006 | 0.014868 | 0.660863 |
| Sede | 0.5131 | 0.4609 | 0.0211 | 0.0023 | 0.0009 | 0.0005 | 0.0011 | 0.00871 | 0.602638 |

the previous section: the lower-frequency regions (below 0.09) are predominantly predicted by the higher-dimensional hypercomplex layers, effectively capturing global trend patterns; whereas the higher-frequency regions are predicted by the lower-dimensional complex and real-valued layers, capturing localized seasonal signals. And because the model is channel-independent, this prediction rule does not change with the channel.

However, a notable divergence from conventional decomposition design emerged (cite). Traditional modeling intuition often assumes that low-frequency components are more difficult to predict than high-frequency components, leading to designs that allocate greater model capacity to handle low-frequency signals. Contrarily, our model, after training, reveals an opposite pattern: the complex and hypercomplex MLP, which constitute the majority of the model's parameters, primarily focus their predictive capacity on a specific low-frequency band—particularly the interval $[0.04 − 0.09]$—while selectively predicting auxiliary low-frequency components in $[0 − 0.01]$, $[0.01 − 0.04]$, and $[0.09 − 0.14]$. Surprisingly, the Real MLP, despite having the fewest parameters, is responsible for predicting the complex high-frequency components.

Tables 14, 15, and 16 summarize the quantitative results of our analyses on the ETTm1 and ETTm2 datasets, both of which exhibit lower frequency characteristics compared to the ETTh1 dataset.

When comparing across datasets, we observe that the low-frequency nature of ETTm1 and ETTm2, coupled with their training challenges, results in the Real MLP struggling to capture prominent high-dimensional features. Consequently, the predicted signals from the MLP tend to concentrate in lower frequency ranges overall.

Nonetheless, a closer examination of the frequency distribution across low-frequency intervals, along with metrics such as the Mean Amplitude Frequency (MAF) and Pearson correlation coefficient, reveals a clear stratification pattern in the predictive frequencies of the different MLP. Despite operating predominantly at low frequencies, the predictions from each RHR-MLP exhibit distinct characteristics that reflect the integration of diverse feature sources.

These findings suggest that the decomposition mechanism learned by the model is indeed functioning as intended, effectively disentangling components with different frequency properties. However, the

Table 16: Mean PVE per RHR-MLP per frequency bin on channel 6 of ETTm2, with corresponding Mean Absolute Frequency (MAF) and Pearson correlation coefficient.

| | [0.00-0.01] | [0.01-0.04] | [0.04-0.09] | [0.09-0.14] | [0.14-0.20] | [0.20-0.27] | [0.27-0.50] | MAF | Pearson |
|---|---|---|---|---|---|---|---|---|---|
| Real | 0.1545 | 0.7596 | 0.0506 | 0.0326 | 0.002 | 0.0004 | 0.0003 | 0.017574 | 0.016364 |
| Comp | 0.9773 | 0.0108 | 0.0043 | 0.0027 | 0.0015 | 0.001 | 0.0023 | 0.002114 | 0.055557 |
| Quat | 0.3066 | 0.4821 | 0.1502 | 0.0454 | 0.005 | 0.0031 | 0.0075 | 0.026367 | 0.096686 |
| Octo | 0.0671 | 0.872 | 0.0454 | 0.0046 | 0.0033 | 0.0023 | 0.0053 | 0.017973 | 0.315934 |
| Sede | 0.9916 | 0.0068 | 0.0008 | 0.0004 | 0.0001 | 0.0001 | 0.0002 | 0.000295 | 0.169416 |

results also indicate that there is potential for further improvement and optimization to expand the decomposition frequency range and improve the prediction performance.

## L  WHY THE METHOD WORKS: RMT AND HYPERCOMPLEX–SPECTRAL PERSPECTIVES

### L.1  J.1 RANDOM–MATRIX PERSPECTIVE: EFFECTIVE–RANK COLLAPSE AND SPECTRAL BIAS IN HYPERCOMPLEX LAYERS

Let $x \in \mathbb{R}^d$ be embedded into a hypercomplex algebra, quaternions $\mathbb{H}$ or octonions $\mathbb{O}$, by identifying $\mathbb{R}$ with the zero–imaginary subfield and applying a random hypercomplex weight $W \in \mathbb{H}^{m \times d}$ or $W \in \mathbb{O}^{m \times d}$. Via standard real block representations, this induces a real linear map

Although $\mathcal{R}(W)$ is generically full rank and has *more* real parameters than a comparable real/complex layer, its *effective* number of modes is smaller.

**Key claim (stable rank reduction).** Define the stable rank

$$\text{sr}\left(\mathcal{R}(W)\right) ; =; \frac{|\mathcal{R}(W)|F^2}{|\mathcal{R}(W)|2^2}. \tag{15}$$

Relative to real/complex random matrices, hypercomplex ensembles exhibit symmetry–induced eigenvalue/singular–value degeneracies, yielding repeated eigenvalues and concentrated spectra . This is speculated from symmetry-induced degeneracies in simple cases: when N=2, quaternions show 2- to 4-fold eigenvalue repetitions (aligned with Dyson ensemble $\beta = 4$), while octonions display up to 8-fold degeneracy ($\beta = 8$)(Forrester, 2017). Let $\sigma_i$ be the singular values and $p_i = \sigma_i^2/|\mathcal{R}(W)|F^2$. The spectral entropy

$$H(\sigma); =; -\sum i p_i \log p_i \tag{16}$$

is reduced by these degeneracies, and the ratio in equation 15 correspondingly decreases as $|\mathcal{R}(W)|_2$ inflates relative to $|\mathcal{R}(W)|_F$.

Lower stable rank means optimization gravitates toward solutions with a few dominant singular directions. Functionally, the layer acts as a *low–pass* operator: it favors smooth, low–frequency structure while attenuating high–frequency oscillations/noise, consistent with implicit/spectral low–rank biases observed in over–parameterized linear networks (Fridovich-Keil et al., 2022; Rahaman et al., 2019).

In noisy time series, truncating high frequencies reduces variance. The algebra–induced rank collapse thus serves as a built–in regularizer: despite large parameter counts, admissible mode diversity is curtailed by symmetry, improving stability against outliers and measurement noise.

### L.2  J.2 HYPERCOMPLEX–SPECTRAL PERSPECTIVE: QFT/OFT PROPERTIES AS INDUCTIVE BIAS FOR MULTIVARIATE TIME SERIES

The quaternion Fourier transform (QFT) (Ell, 1992; Cheng & Kou, 2018; Sangwine, 1996) and octonion Fourier transform (OFT) (Hahn & Snopek, 2011; Baszczyk & Snopek, 2017) extend the complex Fourier transform to vector–valued signals while preserving core theorems:

- **Hermitian symmetry.** For real inputs, spectra are conjugate–symmetric, eliminating redundant frequencies without loss (Salehi et al., 2013). In hypercomplex embeddings, this symmetry extends across channels, reducing spectral redundancy in multichannel data.

- **Plancherel–Parseval.** Energy is preserved between time and frequency domains, supporting power–preserving linear operators and interpretable orthogonal decompositions (Alessio, 2016).

- **Wiener–Khintchine.** Autocorrelation equals the inverse transform of the power spectral density, now in the hypercomplex domain, exposing both intra–channel and cross–channel dependencies (Zbilut & Marwan, 2008).

Windowed variants (e.g., windowed OFT) provide time–frequency localization while retaining these guarantees, improving detection of transient, multiscale structure (Bhat & Sheikh, 2023).

**Why this helps modeling.**

1. *Phase–aware coupling.* Hypercomplex units encode amplitudes and *joint* phases/orientations via non–commutative products; the induced real block operator performs coupled rotations/scalings across channels, enabling orientation–aware filtering that scalar models cannot match at the same parameter budget.

2. *Energy– and correlation–preserving structure.* Because QFT/OFT preserve energy and connect spectra to autocorrelation, hypercomplex–constrained linear maps behave as phase–coherent, power–preserving filters, stabilizing optimization and aiding interpretability.

3. *Built–in multichannel priors.* Algebraic constraints impose cross–channel couplings (akin to Cauchy–Riemann–type relations), pruning spurious degrees of freedom and sharpening generalization for vector–valued time series.

The spectral theorems specify *what* structure is preserved (energy, symmetry, correlations); the RMT view explains *how* algebra enforces a compact dominant–mode set (implicit low rank). Together, they imply a phase–coherent, low–variance estimator that captures trends and stable cross–channel relations while suppressing noise—matching empirical behavior.

**Summary.** Hypercomplex representations provide (i) an *implicit low–rank, low–pass bias* via spectral degeneracies (RMT), and (ii) the *right spectral invariants* (Hermitian symmetry, Plancherel–Parseval, Wiener–Khintchine) for multichannel time series (QFT/OFT). This combination explains why our method is data–efficient, noise–robust, and strong on long–horizon trends—even when nominal parameter counts are large.

## M    SHOW CASE

In this section, we present the complete results of the case study. For each dataset, we randomly selected 2–3 channels from the first seven channels for visualization. The detailed outcomes are summarized as follows. While the predictive performance and visualizations appear broadly similar across models—reflecting the strong overall baseline performance—our model consistently demonstrates more accurate predictions of low-frequency components, particularly evident in the weather dataset and other representative cases.

## N    LIMITATIONS

Our method introduces a novel time series forecasting model leveraging multiple hypercomplex spaces, but it also faces several limitations. First, PyTorch natively supports only real and complex numbers, so all hypercomplex data structures, linear layers, and activation functions had to be implemented from scratch. Computations are simulated in the real domain, leading to slower training and higher memory demands—e.g., a sedenion layer consumes 16× more memory than its real counterpart. Second, optimizers such as Adam are not inherently designed for hypercomplex arithmetic, resulting in slower convergence and increased training fluctuations. Third, despite our efficient implementation, memory usage and runtime scale unfavorably with dimensionality due to the lack of specialized hardware or data structures. Beyond hardware constraints, our approach is also limited in representation design. Current practice relies mainly on zero-padding to extend real inputs and Cayley–Dickson constructions to define hypercomplex spaces, but alternative transformations into different hypercomplex spaces remain underexplored. It is unclear whether other structured spaces or mappings might yield stronger inductive biases for time series, and these directions require substantial theoretical investigation. Finally, advanced transformations, such as generalized hypercomplex Fourier methods, are not yet implemented due to both theoretical and computational barriers. We remain optimistic that future work—developing new transformation schemes, theoretical foundations, and hardware/optimizer support—will greatly enhance the practicality and performance of hypercomplex models.

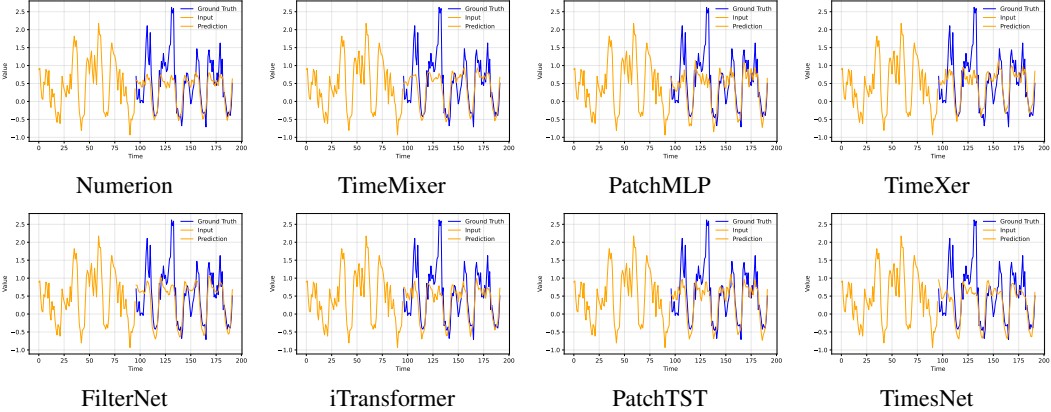

Figure 31: ETTh1 Channel 1

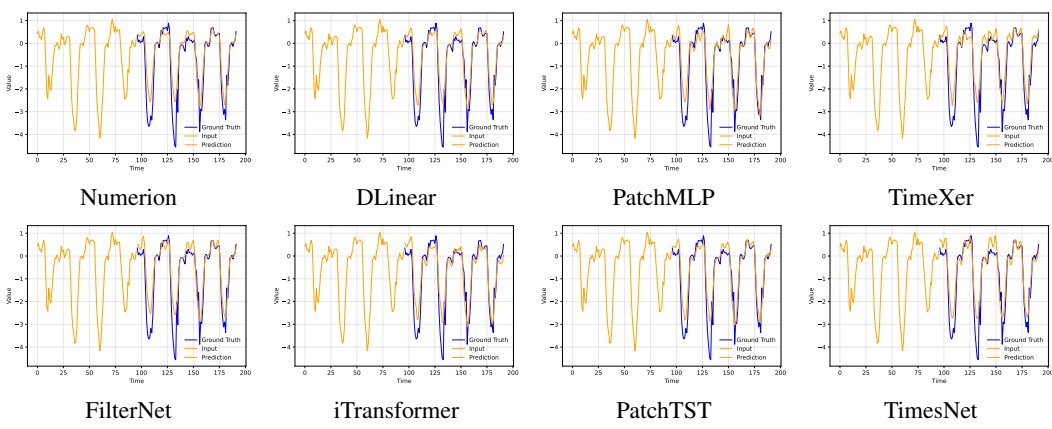

Figure 32: ETTh1 Channel 0

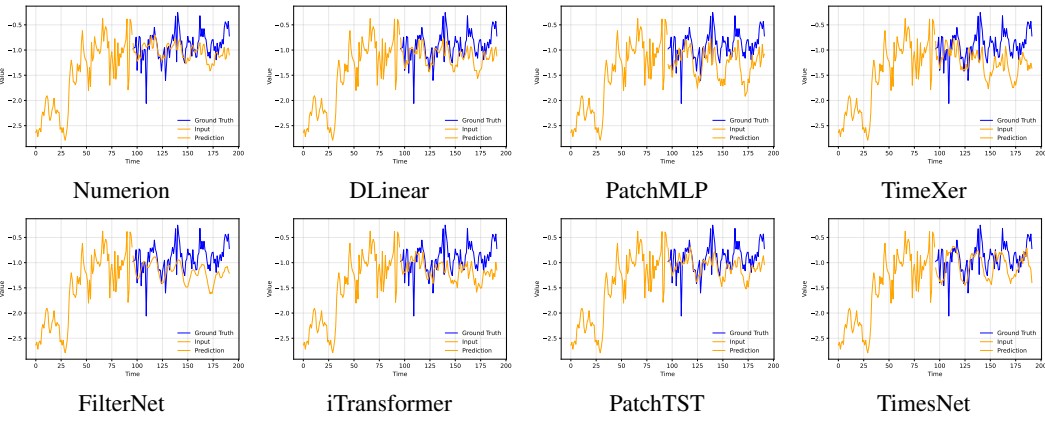

Figure 33: ETTh2 Channel 0

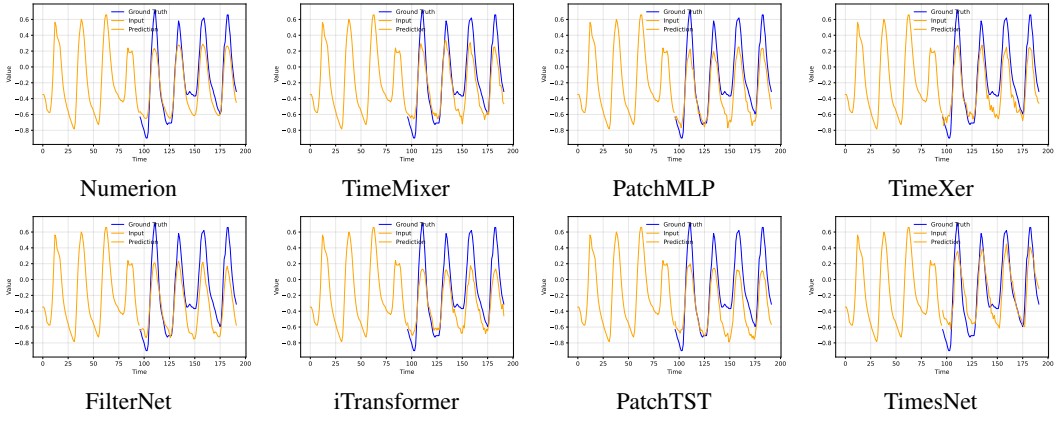

Figure 34: ETTh2 Channel 6

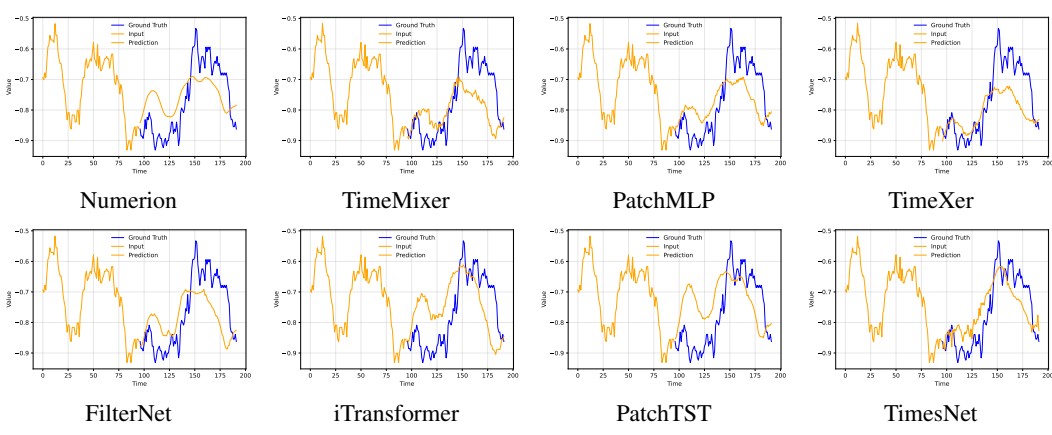

Figure 35: ETTm1 Channel 6

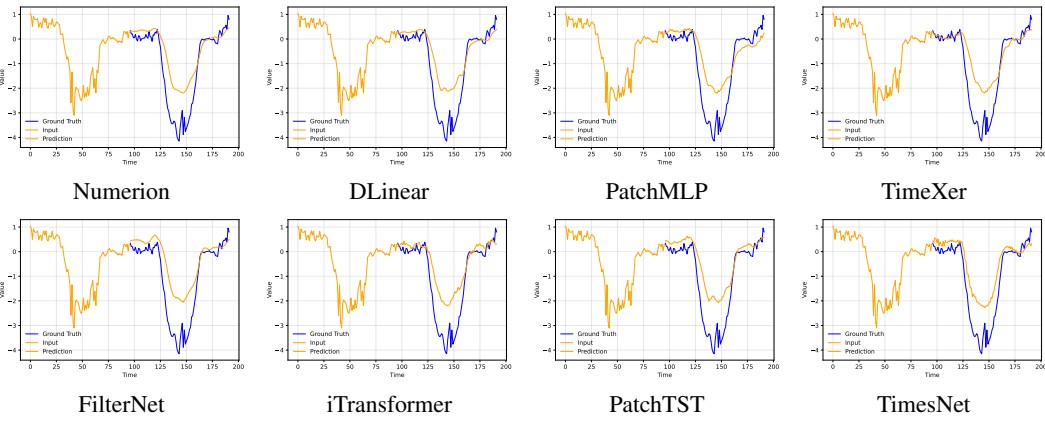

Figure 36: ETTm1 Channel 0

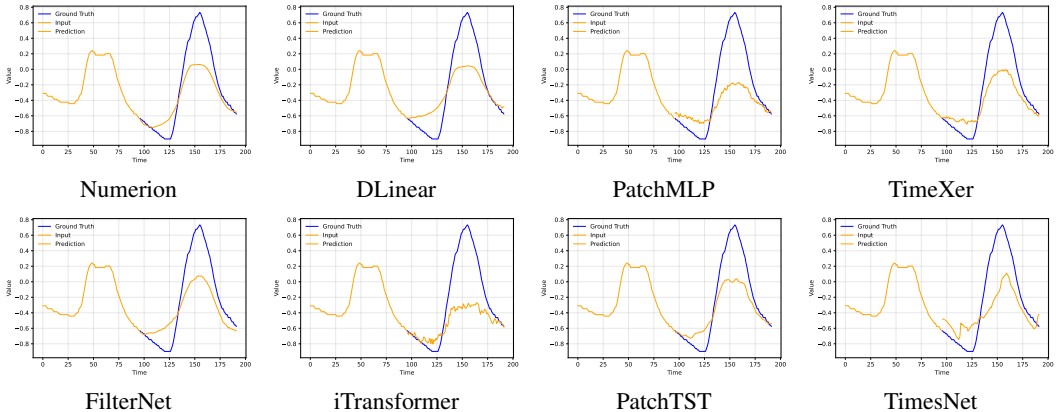

Figure 37: ETTm2 Channel 6

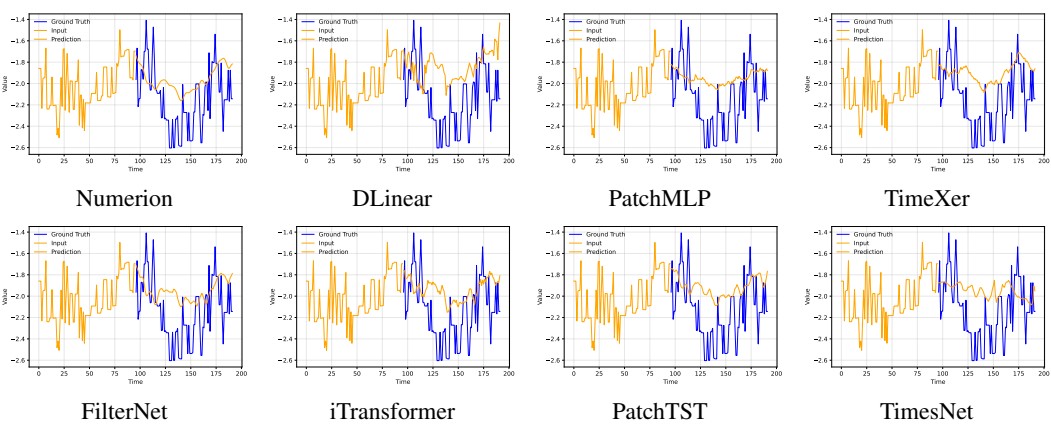

Figure 38: ETTm2 Channel 4

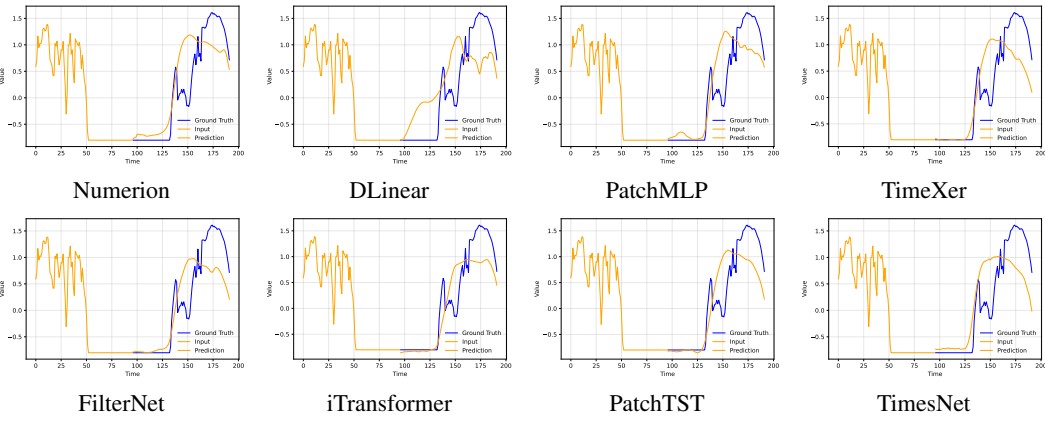

Figure 39: Solar Energy Channel 0

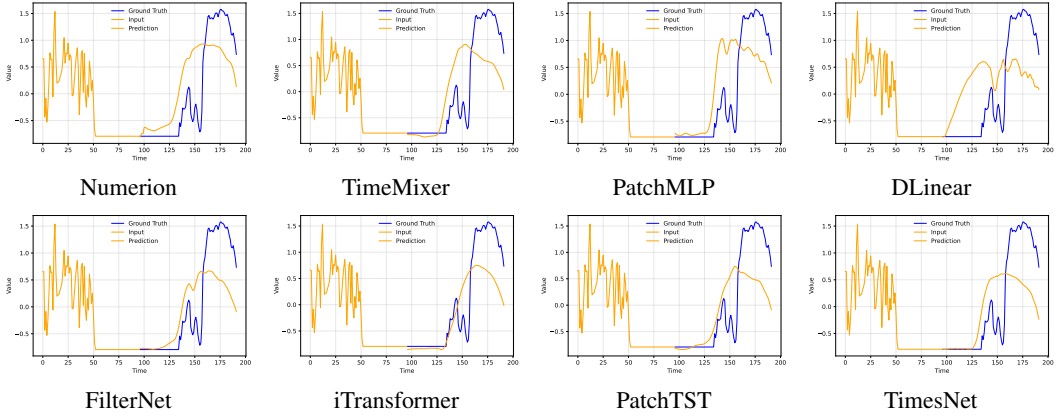

Figure 40: Solar Energy Channel 4

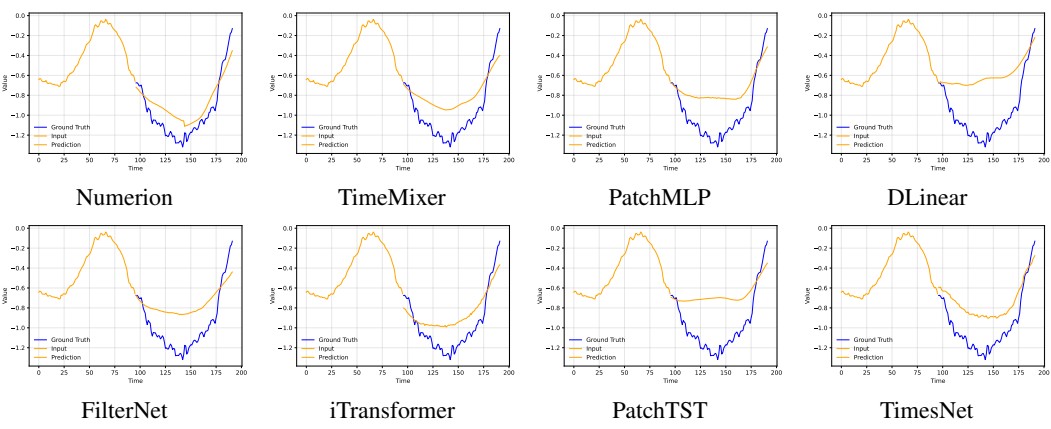

Figure 41: Weather Channel 2

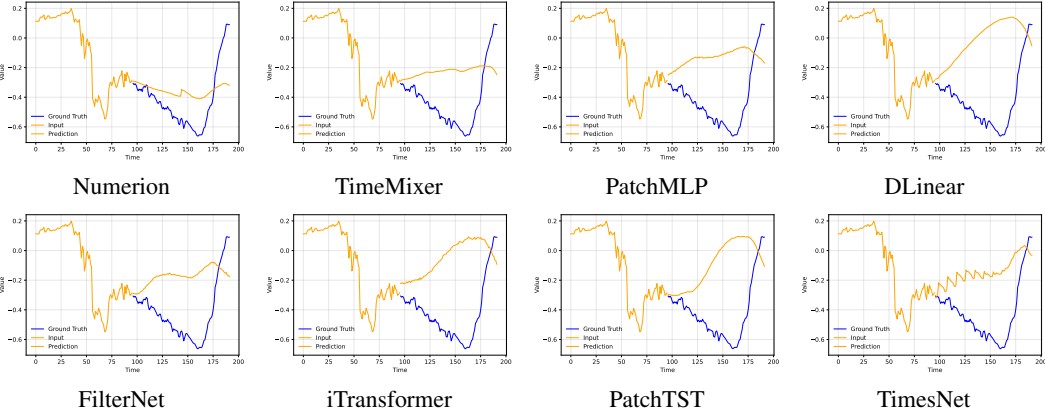

Figure 42: Weather Channel 3

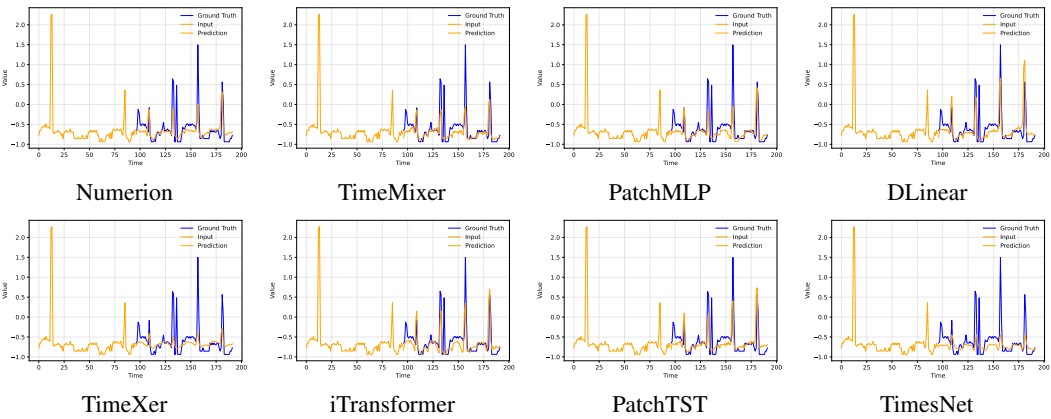

Figure 43: Electricity Channel 0

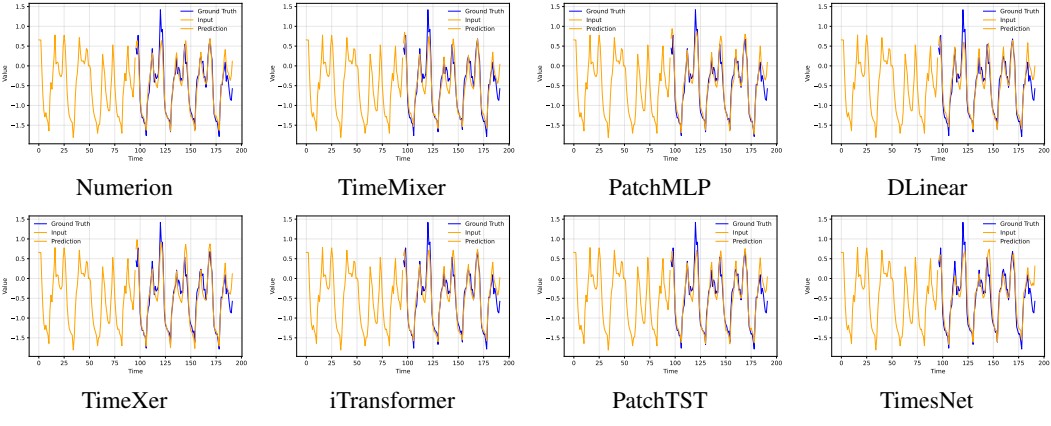

Figure 44: Electricity Channel 5

