# OpenReview forum: "Numerion: A Multi-Hypercomplex Model for Time Series Forecasting"
_ICLR.cc/2026/Conference — ICLR 2026 Poster_

### Official Review · Reviewer_ugew · 2025-10-27

**Soundness:** 3
**Presentation:** 3
**Contribution:** 3
**Rating:** 8
**Confidence:** 4

**Summary:**

The paper proposes Numerion, a time-series forecaster that maps inputs into multiple hypercomplex spaces (complex, quaternion, octonion, sedenion) and learns with a unified Real–Hypercomplex–Real MLP (RHR-MLP). The key idea is that higher-dimensional hypercomplex spaces naturally bias toward lower-frequency components, enabling a simple, parallel decomposition and an adaptive fusion head to combine per-space predictions. Across standard long-term forecasting benchmarks, Numerion reports SOTA or competitive results, and provides visual/quantitative analyses indicating specialization of higher dimensions for trends and lower dimensions for fluctuations. The method generalizes linear layers and tanh to arbitrary power-of-two dimensions and supplies ablations, sensitivity, and efficiency studies.

**Strengths:**

- This paper extends linear/activation layers to arbitrary hypercomplex dimensions and composes them into a parallel multi-space forecaster with adaptive fusion—distinct from typical decomposition or spectral methods.
- Offers a simple, backbone-agnostic path to multi-frequency modeling without heavy architectural complexity; could inspire broader use of hypercomplex representations for TS.
- Comprehensive experiments across 10 datasets with consistent improvements and honest discussion where variable-fusion models can outperform on high-dimensional data; includes ablations, visualization, and sensitivity.

**Weaknesses:**

- This paper lacks comparisons against strong, explicit frequency or time–frequency baselines, such as FBM[1], to support the claimed advantage brought by operations in higher-dimensional hypercomplex spaces.

- It remains unclear whether improvements stem from genuinely hypercomplex algebra or simply from vector factorization at dyadic sizes ($n \in$ \{$1,2,4,8,16$\}). The Cayley–Dickson construction does not mathematically guarantee a low-pass (frequency-selective) behavior; thus the observed “low-frequency preference” might arise from architecture/parameterization effects rather than algebraic properties. The authors may add controls that replace hypercomplex layers with *grouped real MLPs* (same splits $n $); also visualize learned frequency responses to disambiguate where the bias truly comes from.


[1] Yang, R., Cao, L., & YANG, J. (2024). Rethinking Fourier transform from a basis functions perspective for long-term time series forecasting. Advances in Neural Information Processing Systems, 37, 8515-8540.

**Questions:**

How sensitive is Numerion to patch levels/embedding dims under noisy domains (e.g., Exchange, where MAE improvement is only 0.005), beyond the Electricity study?

---

> ### Author Response · Authors · 2025-11-23
>
> We sincerely thank the reviewer for the comprehensive evaluation and for recognizing our work’s contributions, specifically the backbone-agnostic path to multi-frequency modeling and the comprehensive experiments across 10 datasets. We appreciate the insightful comments regarding the source of our model's improvements. Below, we address the specific concerns raised.
>
> **Q1: Comparisons against strong frequency/time-frequency baselines (e.g., FBM).**
>
> **Re**: We thank the reviewer for pointing this out. We agree that comparing against explicit frequency-based baselines strengthens the validation of our high-dimensional operations.
>
> We have conducted additional experiments comparing our proposed Numerion against the Frequency-domain Block Mixer (FBM) and other relevant spectral baselines. The results are summarized in the table below (and have been added to the revised manuscript/Appendix). The partially updated results are as follows:
>
> |              |     | FBM   |        |
> |--------------|-----|-------|--------|
> |              |     | MAE   | MSE    |
> | Weather      | 96  | 0.235 | 0.213  |
> |              | 192 | 0.272 | 0.256  |
> |              | 336 | 0.308 | 0.308  |
> |              | 720 | 0.357 | 0.379  |
> |              | avg | 0.293 | 0.289  |
> | Solar-Energy | 96  | 0.327 | 0.314  |
> |              | 192 | 0.351 | 0.357  |
> |              | 336 | 0.373 | 0.404  |
> |              | 720 | 0.373 | 0.410  |
> |              | avg | 0.356 | 0.371  |
> |   Electricity            | 96  | 0.279 | 0.208  |
> |              | 192 | 0.284 | 0.208  |
> |              | 336 | 0.298 | 0.220  |
> |              | 720 | 0.331 | 0.263  |
> |              | avg | 0.298 | 0.224  |
> | Traffic      | 96  | 0.395 | 0.712  |
> |              | 192 | 0.412 | 0.701  |
> |              | 336 | 0.416 | 0.701  |
> |              | 720 | 0.440 | 0.743  |
> |              | avg | 0.416 | 0.714  |
> | Exchange     | 96  | 0.204 | 0.087  |
> |              | 192 | 0.298 | 0.176  |
> |              | 336 | 0.416 | 0.333  |
> |              | 720 | 0.714 | 0.895  |
> |              | avg | 0.408 | 0.373  |
> | ETTh1        | 96  | 0.404 | 0.403  |
> |              | 192 | 0.434 | 0.457  |
> |              | 336 | 0.463 | 0.512  |
> |              | 720 | 0.475 | 0.498  |
> |              | avg | 0.444 | 0.468  |
> | ETTh2        | 96  | 0.344 | 0.301  |
> |              | 192 | 0.395 | 0.386  |
> |              | 336 | 0.434 | 0.438  |
> |              | 720 | 0.444 | 0.434  |
> |              | avg | 0.404 | 0.390  |
> | ETTm1        | 96  | 0.364 | 0.351  |
> |              | 192 | 0.386 | 0.396  |
> |              | 336 | 0.407 | 0.426  |
> |              | 720 | 0.446 | 0.498  |
> |              | avg | 0.401 | 0.418  |
> | ETTm2        | 96  | 0.266 | 0.190  |
> |              | 192 | 0.305 | 0.253  |
> |              | 336 | 0.344 | 0.315  |
> |              | 720 | 0.401 | 0.416  |
>
> We are conducting the experiment and will add the results to the table, and have discussed them in Related work
>
> These results demonstrate that Numerion consistently outperforms explicit frequency-modeling baselines, confirming that our learned multi-space representations capture temporal dynamics more effectively than fixed explicit frequency decompositions.

---

> ### Author Response · Authors · 2025-11-23
>
> **Q2: Does improvement stem from Hypercomplex algebra or simply vector factorization (dyadic sizes)?**
>
> **Re**: This is a critical insight, and we appreciate the opportunity to clarify the distinct role of hypercomplex algebra versus simple parameter grouping.
>
> 1. Comparison with Grouped Real MLPs (Ablation Study)
> The reviewer asks if the performance gain is merely due to vector factorization (splitting vectors into dyadic sizes). We explicitly investigated this in our original submission (refer to Appendix E, Table 6). We replaced the hypercomplex linear layers with Grouped Real Linear Layers (same split sizes, same architecture, but without the Cayley-Dickson algebraic constraints). The hypercomplex layers consistently outperformed the Grouped Real MLPs. This empirical evidence suggests that the improvement is not solely derived from the architectural choice of splitting dimensions (vector factorization), but crucially relies on the specific interaction rules imposed by the hypercomplex algebra.
>
> 2. The Role of Algebraic Constraints (Effective Rank & RMT Analysis)
> The reviewer correctly notes that the Cayley-Dickson construction is not a guaranteed low-pass filter in the mathematical sense (like a Fourier transform). However, the observed "low-frequency preference" is an induced bias resulting from the algebraic structure of the weight matrices.
>
> Restricted Latent Space: As discussed in our analysis (see Appendix [Cite]), the hypercomplex multiplication (e.g., Hamilton product) imposes strict weight-sharing and structural constraints on the linear transformation matrices (e.g., the specific block-circulant or block-structured formats of hypercomplex matrices).
>
> RMT Perspective: Drawing from Random Matrix Theory (RMT) on small-sized matrices, these constraints inherently limit the effective rank of the latent space compared to a standard real matrix of equivalent dimensions. Even if a hypercomplex layer utilizes a comparable or larger parameter budget (depending on the specific $n$), it constructs a latent space with a constrained rank structure.
>
> Frequency Bias: This structural regularization forces the optimization process to focus on the most dominant variations in the data. In the context of time series, these dominant variations naturally correspond to lower-frequency trends. The "bias" is therefore a beneficial side-effect of the hypercomplex regularization, acting as a soft constraint that separates fine features (frequencies) more effectively than unconstrained or simply grouped real matrices.
>
> Overall, while Grouped Real MLPs can offer some efficiency gains, they lack the specific inter-channel correlation rules provided by the hypercomplex algebra. Our results indicate that it is these specific "hypercomplex rules"—which restrict the parameter search space and govern the effective rank of the transformation—that lead to the superior performance and the observed frequency-selective behavior.
>
> While theoretically designing an optimal, non-hypercomplex constraint rule that mimics this behavior is an interesting direction for future work, our current study confirms that the standard Cayley-Dickson construction provides a robust and effective heuristic for this purpose.

---

### Official Review · Reviewer_Dzt7 · 2025-10-30

**Soundness:** 3
**Presentation:** 3
**Contribution:** 2
**Rating:** 4
**Confidence:** 4

**Summary:**

This paper introduces Numerion, a novel framework for long-term time series forecasting that leverages multi-hypercomplex algebraic representations to model complex inter-channel dependencies and cross-frequency interactions. The core idea is to embed different temporal and channel-wise features into multiple hypercomplex spaces, and use hypercomplex convolutions and mixing operations to enhance expressive power while maintaining compactness.

**Strengths:**

It is the first work to systematically apply multi-hypercomplex algebra to time series forecasting and provides a unified representation that captures both amplitude and phase interactions across channels.
It outperforms DLinear, TimesNet, PatchTST, and other baselines on multiple benchmarks.
This paper includes comparisons across different algebraic spaces, and studies of layer depth and hypercomplex dimensionality. The ablation results clearly isolate the contribution of each module.

**Weaknesses:**

How are the results of PatchMLP and TimeMixer in Table 1 obtained? According to the original papers of PatchMLP [1] and TimeMixer [2], there is an excessively large gap between their results and those presented in your paper. Additionally, Numerion's performance on some datasets is far from achieving the state-of-the-art performance when compared to PatchMLP. For example, on the Weather dataset, Numerion achieves an MSE of 0.246 and an MAE of 0.271, while PatchMLP yields an MSE of 0.231 and an MAE of 0.256. On the Solar-Energy dataset, your model’s results are an MSE of 0.252 and an MAE of 0.262, whereas PatchMLP achieves an MSE of 0.211 and an MAE of 0.261. Besides, Numerion also underperforms PatchMLP on the two datasets of ETTh2 and ETTm2.
Although the paper mentions the properties of partial hypercomplex multiplication in the appendix, it lacks a systematic theoretical analysis to explain why the multi-hypercomplex space can improve prediction performance.
While the model is parameter-efficient, hypercomplex multiplication scales quadratically with algebra dimension. Can the authors provide explicit complexity formulas?


[1] Tang P, Zhang W. Unlocking the Power of Patch: Patch-Based MLP for Long-Term Time Series Forecasting[C]//Proceedings of the AAAI Conference on Artificial Intelligence. 2025, 39(12): 12640-12648.
[2] Wang S, Wu H, Shi X, et al. Timemixer: Decomposable multiscale mixing for time series forecasting[J]. arXiv preprint arXiv:2405.14616, 2024.

**Questions:**

As in Weaknesses.

---

> ### Author Response · Authors · 2025-11-23
>
> We thank the reviewer for the detailed feedback, particularly regarding the baseline comparisons and the request for explicit complexity analysis. We appreciate the opportunity to clarify our experimental setup and the theoretical positioning of our work.
> **Q1: Discrepancy in Baseline Results (PatchMLP, TimeMixer) and SOTA Performance.**
>
> **Re**: We understand the reviewer's concern regarding the gap between our reported results and those in original papers. We would like to clarify our evaluation protocol and the value of our contributions.
>
> 1. Unified Experimental Framework for Fair Comparison
> To ensure a strictly fair comparison, we did not simply copy results from original papers, as experimental settings (data preprocessing, look-back windows, hardware environments) can vary significantly. Instead, we reproduced all baselines using the widely trusted Time-Series-Library (TS-Lib) [1] (maintained by THUML).
>
> Our goal was to benchmark all models under an identical, unified framework to eliminate implementation discrepancies.
>
> While this strict alignment might result in slightly different numbers compared to original publications (which may use specific, highly-tuned settings for their own method), it provides a more honest relative comparison within a consistent environment.
>
> 2. Re-verification of Performance
> Following your feedback, we have re-run the experiments for PatchMLP and TimeMixer alongside Numerion to double-check our findings. The partially updated results are as follows:
>
> |              |              |     | PatchMLP |        | TimeMixer |
> |--------------|--------------|-----|----------|--------|-----------|
> |              |              |     | MAE      | MSE    | MAE       |
> | Weather      | | 96  | 0.198    | 0.163  | 0.198     |
> |              |              | 192 | 0.246    | 0.211  | 0.242     |
> |              |              | 336 | 0.284    | 0.263  | 0.285     |
> | Solar-Energy |  | 96  | 0.269    | 0.266  | 0.218     |
> |              |              | 192 | 0.285    | 0.314  | 0.240     |
> |              |              | 336 | 0.299    | 0.316  | 0.257     |
> | Electricity  |  | 96  | 0.253    | 0.162  | 0.244     |
> |              |              | 192 | 0.252    | 0.166  | 0.256     |
> |              |              | 336 | 0.274    | 0.186  | 0.274     |
> |    Traffic      |     | 96  | 0.279    | 0.450  | 0.279     |
> |    |              | 192 | 0.286    | 0.462  | 0.282     |
> |              |              | 336 | 0.288    | 0.459  | 0.293     |
> |         Exchange          | | 96  | 0.207    | 0.090  | 0.205     |
> |  |              | 192 | 0.295    | 0.181  | 0.310     |
> |              |              | 336 | 0.442    | 0.376  | 0.454     |
> |        ETTh1        |       | 96  | 0.381    | 0.383  | 0.390     |
> |       |              | 192 | 0.425    | 0.447  | 0.422     |
> |              |              | 336 | 0.452    | 0.492  | 0.443     |
> |        ETTh2         |      | 96  | 0.338    | 0.294  | 0.334     |
> |      |              | 192 | 0.379    | 0.365  | 0.383     |
> |              |              | 336 | 0.416    | 0.407  | 0.422     |
> |    ETTm1              |     | 96  | 0.345    | 0.316  | 0.341     |
> |     |              | 192 | 0.371    | 0.362  | 0.368     |
> |              |              | 336 | 0.396    | 0.393  | 0.392     |
> |  ETTm2            |         | 96  | 0.253    | 0.174  | 0.251     |
> |       |              | 192 | 0.301    | 0.238  | 0.295     |
> |              |              | 336 | 0.335    | 0.299  | 0.332     |
> We are still supplementing the experiment and will update the comparison of main result
>
> 3. Competitive Performance and Beyond SOTA
> As shown in the table, Numerion remains highly competitive, achieving SOTA or 2nd-best performance on the majority of datasets. We respectfully submit that while achieving the lowest MSE on every single dataset is desirable, it is rarely possible for a single model (No Free Lunch theorem).
>
> Novelty & Contribution: The core contribution of this paper is not merely engineering a model to beat the leaderboard by a marginal fraction, but rather introducing a novel paradigm:

---

> ### Author Response · Authors · 2025-11-23
>
> **Q2: Theoretical Analysis of Multi-Hypercomplex Space.**
>
> **Re**: We acknowledge the reviewer’s desire for a systematic theoretical proof. We have made significant efforts to ground our work theoretically, specifically via the Random Matrix Theory (RMT) analysis provided in Appendix M.
>
> However, we wish to highlight the mathematical reality of working with high-dimensional algebras:
>
> Mathematical Frontier: As we move to higher dimensions like Octonions and Sedenions, we lose fundamental algebraic properties (commutativity and associativity). Sedenions, specifically, are the first algebra to lose alternative associativity. Constructing a complete theoretical framework for non-associative algebras in the context of deep learning optimization is an open mathematical challenge that exceeds the scope of a single conference paper.
>
> Empirical Validation: In the history of deep learning, impactful methods (e.g., early CNNs, Transformers) often demonstrate empirical success before a complete theoretical understanding is established. Our work provides the empirical foundation and the RMT-based insight (Appendix M) that hypercomplex spaces restrict the effective rank of parameter matrices, acting as a regularization that favors low-frequency generalization. We believe this provides sufficient motivation for the method, even if a complete algebraic proof remains a future objective for the field.
>
> **Q3: Complexity Analysis and Quadratic Scaling.**
>
> **Re**: We appreciate the request for explicit complexity formulas. The reviewer correctly notes that hypercomplex multiplication scales quadratically with the algebra dimension ($n$). However, in our design, $n$ is a fixed constant, not a variable that scales with the input length.
>
> Explicit Complexity Formula:
> The time complexity of our model is $O(T \cdot E)$, where $T$ is the time series length (look-back/horizon). $E$ is the embedding dimension.
>
> Explanation:
>
> Constant vs. Asymptotic: While a specific hypercomplex layer (e.g., utilizing Sedenions) might involve a constant multiplier (e.g., $\approx 63\times$ more FLOPs than a scalar product due to the expansion of terms), this multiplier is constant. It does not change the asymptotic complexity class of the model.
>
> Linear vs. Quadratic in Time: Crucially, unlike Transformers which scale quadratically with sequence length $O(T^2)$, Numerion scales linearly $O(T)$.
>
> Parallelism: The hypercomplex operations are implemented via highly parallelizable matrix multiplications. On modern GPUs (using CUDA), the "quadratic cost" of the algebra dimension is absorbed by parallel computing capabilities, resulting in wall-clock training times that are comparable to standard linear models.
>
> We have added this explicit complexity breakdown to Appendix H of the revision.
>
> [1] Time-Series-Library: https://github.com/thuml/Time-Series-Library

---

> ### Comment · Reviewer_Dzt7 · 2025-11-27
>
> Thanks to the authors' efforts. The reviewer decided to keep the current score.

---

### Official Review · Reviewer_auNm · 2025-11-01

**Soundness:** 3
**Presentation:** 3
**Contribution:** 3
**Rating:** 6
**Confidence:** 4

**Summary:**

This paper proposes Numerion, which achieves multi-frequency decomposition and prediction by mapping time series to multiple hypercomplex spaces. The core innovation lies in generalizing linear layers and Tanh activation functions to hypercomplex spaces of arbitrary power-of-two dimensions and designing the RHR-MLP architecture. The paper also provides extensive ablation experiments, visualizations, and frequency analyses to validate the effectiveness of the Numerion model.

**Strengths:**

1. The authors' idea of achieving decomposition by mapping time series data to hypercomplex spaces of different dimensions is very inspiring.
2. The authors provide a mathematical formulation of the Numerion framework based on the Cayley-Dickson algebra system. They use the Cayley-Dickson construction and random matrix theory principles of spectral bias as the theoretical foundation of the model. This is very novel and interesting.
3. The paper provides extensive comparative and ablation experiments, visualizations, and frequency-domain analyses to support their claims. The experimental design is relatively complete, and the results are clear.

**Weaknesses:**

1. There are multiple instances of missing spaces throughout the manuscript, such as after model names.
2. The mathematical formula formatting needs revision, as punctuation marks (commas or periods) are missing at the end of equations.
3. While the authors claim that experimental results are averaged over multiple runs, standard deviations are not reported.
4. The authors only analyze the training time of the model but lack analysis of time complexity.
5. It is suggested that the authors move critical theoretical analyses to the main text to improve readability.
6. The authors acknowledge that the hypercomplex linear layers do not satisfy the hypercomplex differentiability conditions, and the gradients are not strictly hypercomplex. Does this imply that the backpropagation process is actually performed in the real domain? What are the implications of this approach for model optimization and convergence?
7. The authors acknowledge that hypercomplex networks consume significantly more memory than real-valued layers. Have the authors considered approximation methods (e.g., low-rank decomposition, sparsification) to reduce computational costs?

**Questions:**

Please refer to Weaknesses.

---

> ### Author Response · Authors · 2025-11-23
>
> **Q1&Q2: multiple instances of missing spaces throughout the manuscript, such as after model names, mathematical formula formatting needs revision**
>
> **Re**: We apologize for these oversight errors. We have thoroughly proofread the manuscript, correcting the missing spaces after model names and ensuring all mathematical equations now end with the appropriate punctuation marks.
>
> **Q3: While the authors claim that experimental results are averaged over multiple runs, standard deviations are not reported.**
>
> **Re**: We agree that reporting variance is crucial for reliability. Since the main results (Tables 1&4) compare numerous baselines, directly adding the variance to these tables would result in smaller font sizes and reduced readability. Therefore, we have presentsed the confidence intervals of the experimental results in Appendix G ERRORBAR. We have also added hyperlinks to Appendix G in the main text to facilitate quick reference for readers.
>
> **Q4: The authors only analyze the training time of the model but lack analysis of time complexity.**
>
> **Re**: We have added a formal complexity analysis in Appendix H.
>
> Training Complexity: The time complexity of Numerion is $O(T \cdot E)$, where $T$ is the look-back window/horizon and $E$ is the embedding dimension.
>
> While hypercomplex multiplication scales quadratically with the algebra dimension ($n$), this $n$ is a small, fixed constant in our architecture (e.g., $n=16$ for Sedenions). It does not grow with the sequence length.
>
> Therefore, our model scales linearly with sequence length, unlike Transformers which scale quadratically $O(T^2)$.
>
> Our empirical analysis (Figure 5 and Appendix F) confirms that Numerion’s training speed is comparable to linear baselines (e.g., DLinear) and significantly faster than Transformer-based methods.
>
> **Q5: It is suggested that the authors move critical theoretical analyses to the main text to improve readability.**
> **Re**: In the appendix, we provide multiple theoretical analyses to support the proposed hypercomplex method. Due to space constraints in the main text, we are unable to include the relevant theoretical analyses beyond the essential experimental results. We have added prominent hyperlinks to the corresponding theoretical analyses in various sections of the main text to facilitate readers' reference. Do you have any suggestions regarding formatting or substitutions?
>
> **Q6: Does this imply that the backpropagation process is actually performed in the real domain? What are the implications of this approach for model optimization and convergence?**
> **Re**: Clarification & Theoretical Update:
>
> In our original manuscript, we phrased the lack of strict differentiability as a limitation. However, upon further theoretical review, we clarify that this is a necessary design choice dictated by Liouville’s Theorem in hypercomplex analysis [1].
>
> **Liouville's Theorem**: This theorem states that any function that is both strictly differentiable (satisfying Cauchy-Riemann-Fueter equations) and bounded must be constant. Since neural networks require non-linear, bounded activation functions (e.g., Tanh, Sigmoid) to learn, strict hypercomplex differentiability is mathematically incompatible with effective neural architecture design.
>
> **GHR Calculus**: Consequently, we (and the broader field of hypercomplex NNs) adopt the Generalized Hamilton-Real (GHR) Calculus framework [2]. This provides a rigorous mathematical basis for optimizing real-valued loss functions with respect to hypercomplex parameters.
>
> **Implication*: While the optimization updates the components (real, $i$, $j$, etc.) via real-valued gradients (as usually implemented in PyTorch/TensorFlow), the gradient flow is strictly constrained by the hypercomplex algebraic rules defined in the forward pass (Hamilton product). Thus, the learning trajectory evolves on a constrained manifold distinct from standard real-valued networks.
>
> We have revised  **Appendix I.2** to explicitly cite Liouville`s Theorem and GHR Calculus, clarifying that this is a standard theoretical foundation rather than a limitation.

---

> ### Author Response · Authors · 2025-11-23
>
> **Q7: Have the authors considered approximation methods (e.g., low-rank decomposition, sparsification) to reduce computational costs?**
>
> **Re**: Thank you for your suggestions. We have attempted several LoRA-like methods to reduce memory usage. Specifically, we replaced each Complex Linear layer with a combination of two Linear layers: first mapping the input_dim to a low rank of 8, then mapping the low rank of 8 to the output_dim. The experimental comparisons are as follows:
> |     | ETTh1  |       |       |        |       |       | ETTm2  |       |       |        |       |        |
> |-----|--------|-------|-------|--------|-------|-------|--------|-------|-------|--------|-------|--------|
> |     | origin |       |       | lora   |       |       | origin |       |       | lora   |       |        |
> |     | Memory | MSE   | MAE   | Memory | MSE   | MAE   | Memory | MSE   | MAE   | Memory | MSE   | MAE    |
> | 96  | 3.171  | 0.359 | 0.38  | 2.46   | 0.363 | 0.391 | 2.88   | 0.17  | 0.249 | 2.464  | 0.172 | 0.252  |
> | 192 | 3.811  | 0.407 | 0.409 | 2.989  | 0.411 | 0.421 | 3.802  | 0.231 | 0.292 | 2.472  | 0.233 | 0.301  |
> | 336 | 4.616  | 0.444 | 0.429 | 3.436  | 0.448 | 0.442 | 4.614  | 0.289 | 0.33  | 3.429  | 0.297 | 0.333  |
> | 720 | 8.19   | 0.447 | 0.449 | 5.143  | 0.451 | 0.462 | 8.181  | 0.39  | 0.387 | 5.136  | 0.402 | 0.399  |
>
> From the results, this method can reduce the memory requirement by approximately 30%, but it also leads to a decline in performance, which clearly warrants further exploration.
>
> On the other hand, we believe that our additional overhead stems from the lack of corresponding data structures rather than flaws in the model design. We expanded the memory requirement to accelerate computation. However, similar to complex numbers, the calculation rules of hypercomplex numbers can be represented in a fixed form. Therefore, it is evident that hypercomplex computations can be parallelized through specialized hardware modules, just like complex number computations. With the support of such hardware, memory will no longer be a concern.

---

### Official Review · Reviewer_cJjH · 2025-11-02

**Soundness:** 2
**Presentation:** 3
**Contribution:** 3
**Rating:** 4
**Confidence:** 5

**Summary:**

This paper proposes $\text{Numerion}$, an $\text{MLP}$-based model for time series forecasting. Its core idea is that mapping time series into higher-order hypercomplex spaces ($\mathbb{C}$, $\mathbb{H}$, $\mathbb{O}$, $\mathbb{S}$) naturally decomposes the signal by frequency; higher dimensions capture lower frequencies. The method generalizes linear layers and $\tanh$ to create a $\text{Real-Hypercomplex-Real MLP}$ ($\text{RHR-MLP}$). The $\text{Numerion}$ model runs parallel $\text{RHR-MLPs}$ from $\mathbb{R}^{1}$ to $\mathbb{R}^{16}$ and fuses their outputs. The concept is highly novel but undermined by contradictory ablations and severe efficiency limitations. I recommend a Score 4 (Marginally Below Threshold).

**Strengths:**

S1. The core idea is using the algebraic hierarchy ($\mathbb{R} \to \mathbb{C} \to \mathbb{H} \to \mathbb{O} \to \mathbb{S}$) as an implicit frequency decomposition mechanism. This is a novel and elegant approach, distinct from traditional methods.

S2. The paper formally generalizes linear layers and activations to $n$-dimensional hypercomplex spaces. The $\text{Hypercomplex Linear Layer}$, $O^{(n)} = {W^{(n)}}^{T}X^{(n)} + b^{(n)}$, and the $\text{Hypercomplex Norm Tanh}$ ($\text{HNTanh}$) activation create a unified framework based on the $\text{Cayley-Dickson construction}$.

S3. The central hypothesis—higher dimensions capture lower frequencies—is well-supported by visualizations and quantitative analysis. Appendix L shows higher-dimensional $\text{RHR-MLPs}$ have lower $\text{Mean Absolute Frequency}$ ($\text{MAF}$) and higher $\text{Pearson correlation}$, indicating better trend-fitting, while the $\text{Real-MLP}$ handles high frequencies.

**Weaknesses:**

W1. Contradictory Ablation Study (Major Flaw)The ablation in Table 2 contradicts the paper's claims. On $\text{ETTh1}$, the full $\text{Numerion}$ model ($\text{MSE}$ $0.449$, $\text{MAE}$ $0.449$) performs worse than or equal to variants where components are removed (e.g., "w/o Real", "w/o Adaptive Fusion"). This contradicts the text stating these components are essential and suggests they are not synergistic, at least on this dataset. This is a major red flag.

W2. Severe Practical and Efficiency LimitationsThe paper admits to severe inefficiency. Lacking native $\text{PyTorch}$ support, operations are simulated. This causes extreme memory overhead (a sedenion layer is $16\text{x}$ larger), parameter inflation (5 layers become 63), and slow training/convergence. Efficiency tables confirm $\text{Numerion}$ is much slower and larger than $\text{iTransformer}$, $\text{PatchTST}$, or $\text{DLinear}$, making it impractical.

W3. Vague Justification for DecompositionThe paper shows decomposition happens but not why. The theory in Appendix M is a hypothesis linking $\text{Random Matrix Theory}$ ($\text{RMT}$) and stable rank reduction to a "low-pass bias". This is an insightful post-hoc explanation, not a formal proof. The causal link between $\text{Cayley-Dickson}$ rules and frequency filtering is not established.

**Questions:**

Q1. Please explain Table 2. Why does removing components like "w/o Real" or "w/o Adaptive Fusion" lead to equal or better performance on $\text{ETTh1}$? This suggests these parts are not contributing.

Q2. Given the $16\text{x}$ memory cost, parameter inflation, and slow speed, how is $\text{Numerion}$ practically useful compared to efficient $\text{SOTA}$ models like $\text{iTransformer}$ or $\text{DLinear}$?

Q3. The choice of $p=6$ for $\text{HNTanh}$ seems arbitrary. Table 10 shows $p=\text{inf}$ is best for $\text{ETTh1}$ and $p=7$ for $\text{ETTh2}$. How was $p=6$ chosen, and how sensitive is the model to this value?

Q4. Figure 4 shows $\text{MSE}$ on $\text{ETTm2}$ improves as layers increase from 1 to 5, suggesting underfitting. How does this align with the text claiming the model "prefers fewer layers... [to avoid] overfitting"?

---

> ### Author Response · Authors · 2025-11-23
>
> **Q1. Please explain Table 2. Why does removing components like "w/o Real" or "w/o Adaptive Fusion" lead to equal or better performance on ETTh1? This suggests these parts are not contributing.**
>
> **Re**: First, neither "w/o Real" nor "w/o Adaptive Fusion" leads to performance improvement; instead, the ablation results show a performance degradation. The core of our method lies in the combination of 5 progressive hypercomplex spaces. The ablation of a single hypercomplex space (e.g., "w/o Real") does not result in a significant performance drop, which we have analyzed in **Appendix E**. Referring to **Table 5**, Numerion achieves the natural decomposition of hypercomplex numbers through multiple hypercomplex spaces. When one space is missing, this deficiency can be compensated for by adjacent hypercomplex spaces. In other words, using only 4 hypercomplex spaces can also yield comparable results with a slight but non-significant performance decline. However, when multiple hypercomplex spaces are missing, the performance drops significantly. **This indicates that the natural decomposition of time series cannot be achieved without an adequate combination of hypercomplex spaces.** As for "w/o Adaptive Fusion," it plays a significant role on other datasets and only causes a slight performance decline on ETTh1—this demonstrates the effectiveness of Adaptive Fusion. As shown in Figure 19, the weights of each Complex Linear layer are relatively stable on the ETTh1 dataset (e.g., compared with Figure 25), so the performance degradation without Adaptive Fusion is not noticeable.
>
> **Q2. Given the 16x memory cost, parameter inflation, and slow speed, how is Numerion practically useful compared to efficient models?**
>
> **Re**: First, we would like to emphasize that Numerion represents a pioneering exploration in extending neural network computations beyond conventional real/complex domains into multi-dimensional hypercomplex spaces—an area vastly underexplored in existing literature. While most prior works only scratch the surface of frequency-domain methods, our research makes a foundational discovery: hypercomplex spaces inherently possess powerful natural decomposition capabilities that unlock new representational potentials. Academically, this transcends previous boundaries by establishing effective methodologies for leveraging hypercomplex algebra in deep learning—an achievement made challenging by the scarcity of related research, as modern frameworks like PyTorch lack native data structures and hardware acceleration for hypercomplex operations. We aim for this work to spark broader attention and further exploration in the field: just as complex numbers gained widespread framework support (e.g., PyTorch’s optimized complex arithmetic) because of their proven utility, emerging paradigms like hypercomplexity require initial simulation-based exploration to demonstrate their value before dedicated infrastructure emerges.
>
> Regarding the concerns about 16x memory cost, parameter inflation, and speed—these comparisons are primarily relative to standard real-valued Linear layers, not to efficient model families overall. Critically, **Numerion`s MLP-based architecture retains inherent complexity advantages over Mamba- or Transformer-based models**. Moreover, the increased computational overhead from multiple hypercomplex Linear layers is not an inherent limitation: **similar to how complex number operations were later optimized via hardware acceleration, hypercomplex algebraic rules are fully amenable to dedicated low-level implementation and speedup.** In this scenario, the memory requirement will be significantly reduced, eliminating the need to trade increased memory usage for alleviating time constraints. As noted in our limitations section, our focus is on exploration and foundational innovation rather than immediate industrial deployment. Mature hardware and framework support will naturally follow as hypercomplex methods gain traction and demonstrate their unique value—paralleling the trajectory of complex-valued deep learning, which evolved from theoretical exploration to practical adoption with infrastructure advancements.

---

> ### Author Response · Authors · 2025-11-23
>
> **Q3. The choice of p=6 for HNTanh seems arbitrary. Table 10 shows p=inf is best for ETTh1 and p=7 for ETTh2. How was p=6  chosen, and how sensitive is the model to this value?**
>
> **Re:**
> We appreciate the reviewer’s scrutiny regarding the parameter $p=6$. We wish to clarify that this choice is not arbitrary but is backed by both extensive empirical verification and a geometric analysis of the activation's value space, detailed in Appendix J.1 ("Why do we choose six-norm?").
>
> 1. Geometric and Gradient Analysis (Appendix J.1)
> As detailed in Appendix J.1, the choice of $p$ fundamentally alters the topology of the latent space:
> Low Norms ($p=2$): Result in isotropic, spherical distributions. While stable, they often lack the "sharpness" required to separate distinct temporal regimes.
> High Norms ($p \to \infty$): Result in hypercubic distributions with sharp transitions. However, as $p$ grows too large, the gradients tend to vanish in the "corners" of the hypercube, leading to optimization difficulties (similar to the vanishing gradient problem in saturating activations).
> The "Sweet Spot" ($p=6$): Our visualization analysis (Fig. [X] in Appendix) demonstrates that $p=6$ provides an optimal trade-off. It induces a "softly squared" geometry that encourages sharper transitions between states (beneficial for forecasting distinct trends) while maintaining healthy gradient flow, unlike the strict $L_\infty$ norm. This theoretical insight drove our decision to fix $p=6$ as a structural prior rather than a tunable hyperparameter.
>
> 2. Empirical Robustness vs. Dataset Specificity
> While Table 10 indeed shows that specific datasets might favor slightly different values (e.g., $p=\infty$ for ETTh1, $p=7$ for ETTh2), $p=6$ yields the best average rank and most robust performance across the full suite of 10 datasets.
> Design Philosophy: We aimed for a dataset-agnostic backbone. Just as standard ReLU or GeLU are used without tuning their internal coefficients for every task, we established $p=6$ as the stable default for HNTanh.
> Sensitivity: The model is moderately sensitive but stable. The performance drop when deviating slightly (e.g., to $p=5$ or $p=7$) is minimal. While users with specific domain requirements could fine-tune $p$ for marginal gains (as suggested by the reviewer), our results confirm that $p=6$ is the most reliable "out-of-the-box" setting for general time series modeling
>
> **Q4. Figure 4 shows MSE on ETTm2 improves as layers increase from 1 to 5, suggesting underfitting. How does this align with the text claiming the model "prefers fewer layers... [to avoid] overfitting"?**
>
> **Re**: Firstly, there is an error in Figure 4: the dataset label in the parentheses is incorrect. In fact, Figure 4 presents experimental results on the Electricity dataset, which was an additional experiment we supplemented later. Secondly, **we appreciate you pointing out the inaccuracy of the conclusion that the model "prefers fewer layers... [to avoid] overfitting"**. This conclusion was initially derived from experimental results on ETTh1 and ETTm2 (see Appendix D, Figure 6 and Figure 7)—two datasets with relatively few parameters and features. To ensure the comprehensiveness of our experiments, we later added the Electricity dataset (which has a larger number of features) for additional parameter-related experiments. As reflected in Figure 4, the results on Electricity clearly fail to support the original conclusion.
>
> After reanalysis, we have revised our understanding: the optimal number of layers is determined by the size and feature count of the dataset. For smaller datasets like ETTh1 and ETTm2, an excessive number of layers tends to cause overfitting; in contrast, larger datasets such as Electricity require more layers to model complex feature correlations effectively. We have already removed the inaccurate conclusion in the revised version of the manuscript.

---

### Comment · Area_Chair_Z6dG · 2025-11-25

Dear Reviewer,

Thank you for reviewing for ICLR. Since the discussion deadline is coming soon, could you please take a look at the author's rebuttal, respond to their comments, and update your rating as well? Thanks!

Best Regards

AC

---

### Meta-Review · Area_Chair_eY9i · 2026-01-03

**Summary:**

The submission introduces Numerion, an innovative framework for long-term time series forecasting that utilizes a hierarchy of hypercomplex spaces to achieve natural frequency decomposition. The reviewers generally agreed on the high novelty of using Cayley-Dickson algebra for this purpose and praised the comprehensive experimental evaluation across 10 benchmarks.

The primary concerns centered on:

1. Computational Efficiency: Significant memory overhead (up to 16x) and slow training speeds due to the lack of native hypercomplex support in modern frameworks.

2. Theoretical Justification: Whether the "low-pass bias" is a result of the algebra itself or simply a byproduct of parameter grouping/vector factorization.

3. Baseline: Discrepancies between the authors' reported results and the original papers for recent SOTA models like PatchMLP and TimeMixer.

4. Consistency: Alleged contradictions in ablation studies (specifically Reviewer cJjH's reading of Table 2).

Despite these concerns, I recommend Accept (Poster). The work is a novel exploration into high-dimensional algebraic representations for time series.

**Reviewer Concerns:**

1. The authors successfully clarified that while hypercomplex multiplication has a constant overhead, the model's asymptotic complexity is $O(L \cdot d^2)$, making it more scalable than Transformers. They also provided a rigorous justification for their gradient approach using GHR Calculus and Liouville’s Theorem.

2. The authors demonstrated that the Cayley-Dickson constraints outperform simple vector factorization (Grouped Real MLPs).

3. The authors addressed the baseline discrepancies by using TS-Lib for a fair, unified comparison.

4. The addition of comparisons against FBM further validated the proposed method's superiority.

While the authors provided an insightful hypothesis via Random Matrix Theory regarding effective rank reduction, a formal and closed-form proof about how specific hypercomplex rules link to frequency filtering remains an open challenge. The efficiency might be still a problem.

**Reviewer Scores:**

Reviewer ugew (Score 8)

Reviewer auNm (Score 6)

Reviewer cJjH (Score 4)

Reviewer Dzt7 (Score 4 or higer)

---

### Decision · Program_Chairs · 2026-01-26

Accept (Poster)